# A surge of *p*-values between 0.041 and 0.049 in recent decades (but negative results are increasing rapidly too)

Joost CF de Winter and Dimitra Dodou

Department of BioMechanical Engineering, Delft University of Technology, Delft,
The Netherlands

## ABSTRACT

It is known that statistically significant (positive) results are more likely to be published than non-significant (negative) results. However, it has been unclear whether any increasing prevalence of positive results is stronger in the "softer" disciplines (social sciences) than in the "harder" disciplines (physical sciences), and whether the prevalence of negative results is decreasing over time. Using Scopus, we searched the abstracts of papers published between 1990 and 2013, and measured longitudinal trends of multiple expressions of positive versus negative results, including *p*-values between 0.041 and 0.049 versus *p*-values between 0.051 and 0.059, textual reporting of "significant difference" versus "no significant difference," and the reporting of $p < 0.05$ versus $p > 0.05$. We found no support for a "hierarchy of sciences" with physical sciences at the top and social sciences at the bottom. However, we found large differences in reporting practices between disciplines, with *p*-values between 0.041 and 0.049 over 1990–2013 being 65.7 times more prevalent in the biological sciences than in the physical sciences. The *p*-values near the significance threshold of 0.05 on either side have both increased but with those *p*-values between 0.041 and 0.049 having increased to a greater extent (2013-to-1990 ratio of the percentage of papers = 10.3) than those between 0.051 and 0.059 (ratio = 3.6). Contradictorily, $p < 0.05$ has increased more slowly than $p > 0.05$ (ratios = 1.4 and 4.8, respectively), while the use of "significant difference" has shown only a modest increase compared to "no significant difference" (ratios = 1.5 and 1.1, respectively). We also compared reporting of significance in the United States, Asia, and Europe and found that the results are too inconsistent to draw conclusions on cross-cultural differences in significance reporting. We argue that the observed longitudinal trends are caused by negative factors, such as an increase of questionable research practices, but also by positive factors, such as an increase of quantitative research and structured reporting.

Corresponding author
Joost CF de Winter,
j.c.f.dewinter@tudelft.nl

# INTRODUCTION

## The abundance of positive results in the scientific literature

It is well known that the distribution of *p*-values in the scientific literature exhibits a discontinuity around alpha = 0.05, with *p*-values just below 0.05 being more prevalent

than *p*-values just above 0.05 (*Brodeur et al., 2013*; *Gerber & Malhotra, 2008a*; *Gerber & Malhotra, 2008b*; *Kühberger, Fritz & Scherndl, 2014*; *Pocock et al., 2004*; *Ridley et al., 2007*). The abundance of positive results has often been attributed to selective publication, also called the file drawer effect (*Dwan et al., 2008*; *Hopewell et al., 2009*; *Rothstein, Sutton & Borenstein, 2006*). Another explanation for the discontinuity in the *p*-value distribution is that many statistically significant findings in the literature are false positives. A highly-cited article by *Ioannidis (2005)* claimed that over 50% of the published results that are declared statistically significant are false, meaning that they actually are negative (i.e., null) effects. Concerns regarding publication bias and false positives have been expressed in a variety of research fields, including but not limited to biology and ecology (*Jennions & Møller, 2002*), economics (*Basu & Park, 2014*), medicine and pharmaceutics (*Dwan et al., 2008*; *Hopewell et al., 2009*; *Kyzas, Denaxa-Kyza & Ioannidis, 2007*), neurosciences (*Jennings & Van Horn, 2012*), epidemiology (*Pocock et al., 2004*), cognitive sciences (*Ioannidis et al., 2014*), genetics (*Chabris et al., 2012*; *Ioannidis, 2003*), and psychology (*Ferguson & Heene, 2012*; *Francis, 2013*; *Francis, 2014*; *Franco, Malhotra & Simonovits, 2014*; *Laws, 2013*; *Simmons, Nelson & Simonsohn, 2011*).

False positives can be caused by questionable statistical practices (called '*p*-hacking,' 'fiddling,' 'spin,' or 'data massage') that turn a negative result into a positive result (*Boutron et al., 2010*; *Chan et al., 2014*; *Dwan et al., 2008*; *Kirkham et al., 2010*; *Simmons, Nelson & Simonsohn, 2011*). Selectively removing particular outliers, undisclosed data dredging, selective stopping, incorrect rounding down of *p*-values, or trying out various statistical tests and subsequently reporting only the most significant outcomes are some of the mechanisms that can cause false positives to appear in a publication (e.g., *Bakker & Wicherts, 2014*; *Camfield, Duvendack & Palmer-Jones, 2014*; *Goodman, 2014*; *Leggett et al., 2013*; *Simmons, Nelson & Simonsohn, 2011*; *Strube, 2006*). Yet another mechanism that could cause spurious positive results is data fabrication. A meta-analysis by *Fanelli (2009)* found that 2% of researchers admitted fabricating, falsifying (i.e., distorting), or modifying (i.e., altering) data, and 14% knew a colleague who fabricated, falsified, or modified data. The underlying causes of these phenomena are purportedly an emphasis on productivity (*De Rond & Miller, 2005*), high rejection rates of journals (*Young, Ioannidis & Al-Ubaydli, 2008*), and competitive schemes for funding and promotion (*Joober et al., 2012*). Not just researchers, but also journal editors (*Sterling, Rosenbaum & Weinkam, 1995*; *Thornton & Lee, 2000*) and sponsoring/funding parties (*Djulbegovic et al., 2000*; *Lexchin et al., 2003*; *Sismondo, 2008*) have been criticised for favouring positive results over negative ones. It has been argued that there are so many false positives published in certain fields within the social and medical sciences that these fields are currently in crisis (*Pashler & Harris, 2012*).

As a reaction to the abundance of statistically significant results in the scientific literature, methodologists have emphasized that null results should not remain in the file drawer, and that the decision to publish should be based on methodological soundness rather than novelty or statistical significance (*Asendorpf et al., 2013*; *Dirnagl & Lauritzen, 2010*). Methodologists have also warned of the perils of publishing exploratory research, and have encouraged preregistration of formulated hypotheses, research protocols, and

expected data analyses in public trial registries in an attempt to prevent spurious positive findings (*Asendorpf et al., 2013*; *De Angelis et al., 2004*). Furthermore, statistical corrections have been introduced that *decrease* observed effects, including corrections for publication bias (*Duval & Tweedie, 2000*; *Rücker, Schwarzer & Carpenter, 2008*; *Terrin et al., 2003*) and "credibility calibration" (*Ioannidis, 2008*).

It is worth noting that earlier, in the 1980s and 1990s, the social sciences were also said to be in a crisis. Funding agents threatened to cut budgets because they were frustrated with the ongoing production of small and inconsistent effect sizes (*Hunter & Schmidt, 1996*). As an answer, methodologists introduced several artefact corrections, namely corrections for range restriction, measurement error, and dichotomization. These corrections almost always *increase* the observed effect sizes (*Fern & Monroe, 1996*; *Schmidt & Hunter, 1996*).

## Longitudinal trends of positive versus negative results

Akin to signal detection theory, measures that decrease false positives will lead to more false negatives (*Fiedler, Kutzner & Krueger, 2012*; *Finkel, Eastwick & Reis, in press*). Hence, we ought to ask ourselves whether the alleged crisis reflects the true state of affairs. Have positive results really become more prevalent over the years? Furthermore, it is important to determine whether the prevalence of positive results has dropped recently, which would indicate that the methodological recommendations have been endorsed by the scientific community.

So far, research on longitudinal trends of positive versus negative results has been scarce. An exception is *Fanelli (2012)*, who manually coded 4,656 journal papers. In his article "Negative results are disappearing from most disciplines and countries," Fanelli found that the number of papers providing support for the main hypothesis had increased from 70% in 1990 to 86% in 2007. Fanelli further concluded that the increase was significantly stronger in the social sciences and in some biomedical fields than in the physical sciences. He also reported that Asian countries have been producing more positive results than the United States, which in turn produce more positive results than European countries. It is worth noting that, according to *Fanelli (2012)*, negative results are disappearing *relative* to the number of papers that tested a hypothesis. As clarified by *Fanelli (2014)*: "the *absolute* number of negative results in *Fanelli (2012)* did not show a decline" (italics added).

*Fanelli*'s (*2012*) paper has some limitations. First, although his sample size is impressive (considering that the coding was done manually), the statistical power does not seem high enough for assessing *differences in growth* between disciplines or countries. We extracted and re-analyzed data shown in Fanelli's figures and found that the 95% of the confidence intervals (CI) of the regression slopes are overlapping between disciplines (1.44%/year increase in positive results for the social sciences (95% CI [0.99, 1.90]), 0.92%/year increase for the biological sciences (95% CI [0.53, 1.32]), and 0.65%/year increase for the physical sciences (95% CI [0.19, 1.11]), see also Fig. 1, in which the longitudinal trends in the three scientific disciplines are presented together to assist comparison). *Fanelli* (*2012*, see also *2010*) claimed support for a hierarchy of sciences with physical sciences at the top and social sciences at the bottom, and positive results increasing toward the hierarchy's lower

**Peer**J ______________________________

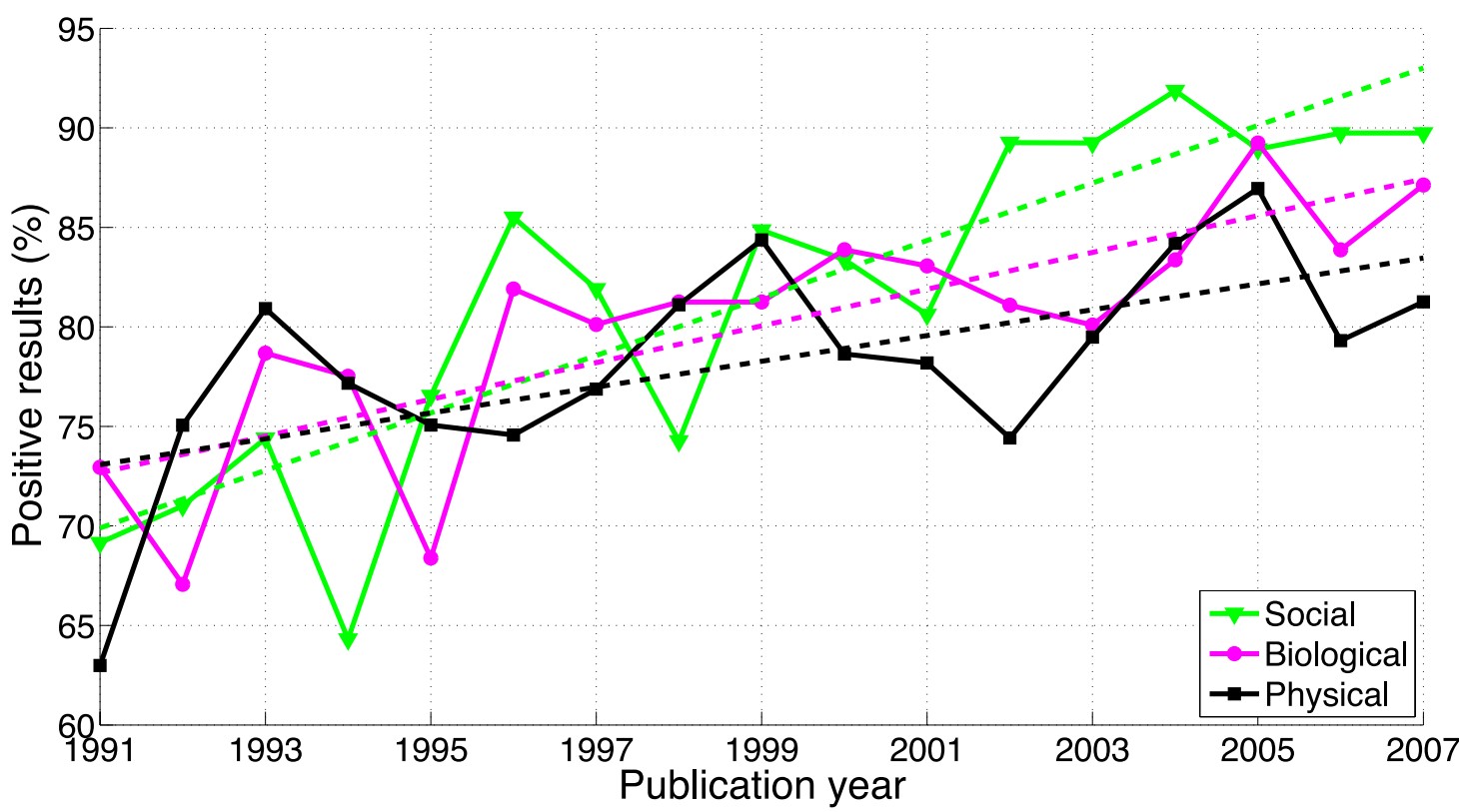

**Figure 1 Number of papers reporting a positive result divided by the total number of papers examined (i.e., papers reporting a positive result + papers reporting a negative result) per publication year, for three scientific disciplines.** The figure was created by graphically extracting the data shown in *Fanelli*'s (*2012*) figures. The dashed lines represent the results of a simple linear regression analysis.

end: "The average frequency of positive results was significantly higher when moving from the physical, to the biological to the social sciences, and in applied versus pure disciplines..., all of which confirms previous findings (*Fanelli, 2010*)" (*Fanelli, 2012*). Although the results shown in Fig. 1 are consistent with this ordinal relationship, the average percentages of positive results over the period 1991–2007 are highly similar for the three disciplines (78.3% in physical sciences [$N = 1{,}044$], 80.1% in biological sciences [$N = 2{,}811$], and 81.5% in social sciences [$N = 801$]). Fanelli's alleged differences between Europe, Asia, and the United States are not statistically strong either. For example, *Fanelli (2012)* reported *p*-values of 0.017 and 0.048 for paired comparisons between the three world regions, based on the results of a stepwise regression procedure that included all main effects and interactions. A second limitation is that *Fanelli*'s (*2012*) assessment of positive/negative results was based on the reading of abstracts or full-texts by the author himself. This approach could have introduced bias, because Fanelli's coding was not blind to the scientific discipline that the papers belong to. Randomization issues are at play as well, because the coding was first done for papers published between 2000 and 2007, and subsequently for papers published between 1990 and 1999.

*Pautasso (2010)* also studied longitudinal trends of positive versus negative research findings. Opposite to *Fanelli*'s (*2012*) manual coding, Pautasso searched automatically

for the phrases "significant difference" versus "no significant difference" (and variants thereof) in the titles and abstracts of papers in the (Social) Science Citation Index for 1991–2008, and in CAB Abstracts and Medline for 1970–2008. *Pautasso (2010)* found that the percentage of both positive and negative results with respect to *all* papers had increased over time (cf. *Fanelli, 2012*, who focused on the percentage of papers reporting a positive result with respect to the selected *papers that tested a hypothesis*, i.e., the sum of papers reporting either positive or negative results). *Pautasso (2010)* further showed that the positive results had increased more rapidly than the negative results, hence yielding a "worsening file drawer problem," consistent with *Fanelli*'s (*2012*) observation that negative results are disappearing. However, some of *Pautasso*'s (*2010*) conclusions seem not in line with *Fanelli*'s (*2010*) hierarchy of sciences. For example, Pautasso found that the worsening file-drawer problem was not apparent for papers retrieved with the keyword "psychol*", whereas psychology lies within the bottom of the hierarchy proposed by *Fanelli (2010)*.

*Leggett et al. (2013)* manually collected 3,701 $p$-values between 0.01 and 0.10 that were reported in all articles published in 1965 and in 2005 in two psychological journals, *Journal of Experimental Psychology: General* (JEP) and *Journal of Personality and Social Psychology* (JPSP). The authors found a distinctive peak of $p$-values just below 0.05. Moreover, the overrepresentation of $p$-values just below the threshold of significance was more manifest in 2005 than in 1965. For example, for *JPSP*, in 2005, there were 3.27 (i.e., 108/33) times as many $p$-values in the range $0.04875 < p \leq 0.05000$ as compared to $p$-values in the range $0.04750 < p \leq 0.04875$, while for 1965 the corresponding ratio was 1.40 (i.e., 28/20; data extracted from Leggett et al.'s figures). Moreover, Leggett et al. found that in 2005, 42% of the 0.05 values were erroneously rounded down $p$-values, as compared to 19% in 1965. According to the authors, these longitudinal trends could be caused by software-driven statisticizing: "Modern statistical software now allows for simple and instantaneous calculations, permitting researchers to monitor their data continuously while collecting it. The ease with which data can be analysed may have rendered practices such as 'optional stopping' (*Wagenmakers, 2007*), selective exclusion of outliers, or the use of post hoc covariates (*Sackett, 1979*) easier to engage in."

*Jager & Leek (2014)* used a parsing algorithm to automatically collect $p$-values from the abstracts of five medical journals (*The Lancet*, *The Journal of the American Medical Association*, *The New England Journal of Medicine*, *The British Medical Journal*, and *The American Journal of Epidemiology*) between 2000 and 2010 and reported that between 75% and 80% of the 5,322 $p$-values they collected were lower than the significance threshold of 0.05. These authors found that the distribution of reported $p$-values has not changed over those years.

All four aforementioned longitudinal analyses require updating, as they cover periods up to 2007 (*Fanelli, 2012*), 2008 (*Pautasso, 2010*), 2005 (*Leggett et al., 2013*), and 2010 (*Jager & Leek, 2014*). Considering the alleged crisis (*Wagenmakers, 2007*) and the exponential growth of scientific literature (*Larsen & Von Ins, 2010*), a modern replication of these works seems warranted. Replication is also needed because *Fanelli*'s (*2012*) work has received

ample citations and attention from the popular press (cf. *Yong, 2012*, stating in *Nature* that psychology and psychiatry are the "worst offenders", based on *Fanelli, 2012*).

### Aim of this research

In summary, it is well known that positive results (i.e., results that are statistically significant) are more likely to be published than negative (i.e., null) results (e.g., *Hopewell et al., 2009*; *Smart, 1964*; *Sterling, Rosenbaum & Weinkam, 1995*). However, it is unclear whether the prevalence of negative results is decreasing over time, whether the increase of positive results is stronger in the "softer" disciplines than in the "harder" disciplines, and whether different regions in the world exhibit different tendencies in reporting positive versus negative results.

The aim of this study was to estimate longitudinal trends of positive versus negative results in the scientific literature, and to compare these trends between disciplines and countries. We chose for an automated search, akin to *Pautasso (2010)* and *Jager & Leek (2014)*. The quantitative analysis of digitized texts, also known as *culturomics*, has become an established method for investigating secular trends in cultural phenomena (e.g., *Michel et al., 2011*).

We investigated longitudinal trends between 1990 and 2013 for $p$-values between 0.041 and 0.049 versus $p$-values between 0.051 and 0.059; that is, for $p$-values on either side of the typical alpha value of 0.05. The assumption is that $p$-values just below the threshold of significance are occasionally the result of questionable statistical practices (cf. *Gadbury & Allison, 2012*; *Gerber & Malhotra, 2008a*; *Masicampo & Lalande, 2012*; *Ridley et al., 2007*). We also investigated longitudinal trends of textual reporting of statistical significance (i.e., "significant difference" versus "no significant difference") to replicate *Pautasso*'s (*2010*) method, as well as of $p < 0.05$ versus $p > 0.05$. A comparison of the reporting of a variety of $p$-values, $p$-value ranges, textual expressions of significance, and idiosyncratic expressions in 1990 versus 2013 (a total of 83 queries) was conducted as well.

## METHODS

### Choice of search engine

Our searches were conducted with Elsevier's Scopus. After trying out other search engines (i.e., Web of Science and Google Scholar), we concluded that Scopus offers the most accurate and powerful search and export features.

Google Scholar indexes more papers than Web of Science and Scopus but has various disadvantages: (1) Google Scholar does not allow for nested Boolean searches; (2) Google Scholar does not provide a possibility for exclusively searching in the abstracts of papers; (3) When the total number of records is greater than 1,000, the estimate of this number is inaccurate[1]; (4) Google Scholar suffers from meta-data errors (*Jacsó, 2008*); (5) Google Scholar suffers from an underrepresentation of work created before the digital age (*De Winter, Zadpoor & Dodou, 2014*); (6) Bulk exportation of records is against the policy of Google Scholar and therefore severely inhibited (*De Winter, Zadpoor & Dodou, 2014*; *Google Scholar, 2014*); and (7) Google Scholar does not provide a possibility for searches

[1] For example, the query "$p = 0.05$" yielded 137,000 results for articles dated 2011, 113,000 results for articles dated 2012, and 96,500 results for articles dated between 2011 and 2012 (search date: 26 November 2014). These findings are logically contradictory.

per discipline or country. Web of Science has various disadvantages as well: (1) Web of Science covers fewer journals and conferences than Scopus; and (2) Web of Science severely underrepresents the social sciences (*De Winter, Zadpoor & Dodou, 2014*).

Scopus has some unique strengths. By using braces ({ }), it allows for searches in which punctuation marks and mathematical operators are taken into consideration, whereas quotation marks are used for more liberal searches. Both Web of Science and Google Scholar neglect punctuation marks and mathematical operators. That is, Scopus is the only one of the three search engines that can distinguish between the search queries {p = 0.001} and {p < 0.001}. Another difference between Scopus and Web of Science (and the associated Essential Science Indicators (*The Thomson Cooperation, 2008*) used by *Fanelli, 2012*) is that, although both indexing services classify journals rather than individual papers, Scopus allows journals to be classified into more than one discipline, whereas Web of Science "assigns each paper to a discipline—and only one discipline" (*Thomson Reuters, 2014*). Results for 'pure' disciplines can be obtained with Scopus by excluding cross-disciplinary papers, whereas cross-classification of records is not possible with Web of Science.

## Longitudinal trends and their comparisons between scientific disciplines and between world regions

Scopus classifies papers into 27 subject areas; we grouped these into three scientific disciplines, each discipline including subject areas as close as possible to *Fanelli*'s (*2012*) classification.

(1) *Social sciences*, consisting of the following Scopus subject areas: Social Sciences; Psychology; Arts and Humanities; Economics, Econometrics and Finance; Decision Sciences; Business, Management and Accounting.
(2) *Biological sciences*, consisting of the following Scopus subject areas: Agricultural and Biological Sciences; Biochemistry, Genetics and Molecular Biology; Medicine; Neuroscience; Immunology and Microbiology; Pharmacology, Toxicology and Pharmaceutics; Veterinary; Environmental Science; Dentistry.
(3) *Physical sciences*, consisting of the following Scopus subject areas: Physics and Astronomy; Engineering; Earth and Planetary Sciences; Chemistry; Chemical Engineering; Materials Science; Energy; Computer Science.

The Mathematics and Multidisciplinary subject areas were not grouped in any of the aforementioned disciplines, to replicate *Fanelli*'s (*2012*) clusters. Health Professions and Nursing were not grouped in any of the disciplines either, because we were not sure into which discipline they should be classified; these two subject areas are relatively small, accounting together for 2.3% (711,406/30,677,779) of all records with an abstract in Scopus with publication date between 1990 and 2013, and it is unlikely that they would have affected our results. Note that we did not explicitly exclude these four subject areas from the searches: an abstract classified into, for example, both Medicine and Nursing would still be classified in the biological sciences.

To investigate whether longitudinal trends in significance reporting differ between regions of the world, we distinguished the following world regions as in *Fanelli (2012)*, namely:

(1) United States (US).

(2) Fifteen European countries (EU15): Austria, Belgium, Denmark, Finland, France, Germany, Greece, Ireland, Italy, Luxembourg, The Netherlands, Portugal, Spain, Sweden, and United Kingdom.

(3) Seven Asian countries (AS7): China, Hong Kong, India, Japan, Singapore, South Korea, and Taiwan.

Note that a paper can belong to multiple world regions due to multiple authors with affiliations in countries from different world regions, or due to an author having multiple affiliations in countries from different world regions.

The following queries were conducted using the advanced search function of Scopus for the abstracts of all papers in the database as well as for each discipline and world region defined above:

(1) ABS({ .}), to extract the total number of papers with an abstract.

(2) $p$-values between 0.041 and 0.049 versus $p$-values between 0.051 and 0.059, namely:
    a. A query containing $p$-values between 0.041 and 0.049, that is: ABS({p = 0.041} OR {p = .041} OR {p = 0.042} OR {p = .042} … OR {p = 0.049} OR {p = .049}), to extract the number of papers with $p$-values between 0.041 and 0.049, that is, just below the typical alpha value of 0.05.
    b. A query containing $p$-values between 0.051 and 0.059, that is: ABS({p = 0.051} OR {p = .051} OR {p = 0.052} OR {p = .052}…OR {p = 0.059} OR {p = .059}), to extract the number of papers with $p$-values between 0.051 and 0.059, that is, just above the alpha value of 0.05.

(3) "Significant difference" versus "no significant difference" and variants thereof, namely:
    a. A query containing typical expressions for reporting significant differences, that is: ABS(({significant difference} OR {significant differences}) AND NOT ({no significant difference} OR {no significant differences} OR {no statistically significant difference} OR {no statistically significant differences})), to extract the number of papers with a textual manifestation of significant results.
    b. A query containing typical expressions for reporting no significant differences, that is: ABS({no significant difference} OR {no significant differences} OR {no statistically significant difference} OR {no statistically significant differences}), to extract the number of papers with a textual manifestation of non-significant results.

(4) $p < 0.05$ (so-called less-than threshold reporting) versus $p > 0.05$, namely:
    a. A query containing variants of the expression $p < 0.05$, that is: ABS({p < 0.05} OR {p < .05} OR {p <= 0.05} OR {p <= .05} OR {p =< 0.05} OR {p =< .05} OR

{p ≤ 0.05} OR {p ≤ .05}), to extract the number of papers reporting significance in reference to the 0.05 threshold of the alpha value.

b. A query containing variants of the expression $p > 0.05$, that is: ABS({p > 0.05} OR {p > .05} OR {p >= 0.05} OR {p >= .05} OR {p => 0.05} OR {p => .05} OR {p ≥ 0.05} OR {p ≥ .05}), to extract the number of papers reporting non-significance in reference to the 0.05 threshold of the alpha value.

The following measures were calculated for items 2–4 of the previous list, per publication year, for all papers in the Scopus database as well as per scientific discipline and per world region:

(1) Percentage of papers reporting significant results ($100\%^\star[S/T]$): the number of papers reporting significant results ($S$) divided by the total number of papers with an abstract ($T$) and multiplied by 100.

(2) Percentage of papers reporting non-significant results ($100\%^\star[NS/T]$): the number of papers reporting non-significant results ($NS$) divided by the total number of papers with an abstract ($T$) and multiplied by 100.

(3) Ratio of significant to non-significant results ($S/NS$): the number of papers reporting significant results ($S$) divided by the number of papers reporting non-significant results ($NS$).

(4) Percentage of significance-testing papers reporting significant results ($100\%^\star[S/(S + NS)]$): the number of papers reporting significant results ($S$) divided by the sum of papers reporting significant and non-significant results ($S + NS$) and multiplied by 100%.

Longitudinal trends were assessed by means of the slope coefficient estimates and corresponding 95% confidence intervals of a simple linear regression analysis, with the publication year as predictor variable and the ratio of significant to non-significant results ($S/NS$) or the percentage of significance-testing papers reporting significant results ($100\%^\star[S/(S + NS)]$) as dependent variable. Note that just-not-overlapping 95% confidence intervals correspond to a $p$-value smaller than 0.006 (*Cumming & Finch, 2005*).

## Pure scientific disciplines and pure world regions

As mentioned above, Scopus classifies papers into multiple subject areas. To gain insight into the number of papers belonging 'purely' to one discipline and to assess how cross-classification may have affected our analyses, we repeated the search queries described in "Longitudinal trends and their comparisons between scientific disciplines and between world regions" for: (1) pure disciplines: papers classified by Scopus into subject areas that belong to the same discipline; and (2) discipline intersections: papers classified by Scopus into subject areas that belong to two or all three disciplines.

Similarly, to gain insight into the number of papers with authors affiliated with countries from the same world region versus the number of papers with author affiliations spanning across world regions, we repeated the search queries described in "Longitudinal

trends and their comparisons between scientific disciplines and between world regions"
for: (1) pure world regions: papers all authors of which were affiliated solely with countries
belonging to one and the same world region; and (2) world region intersections: papers by
authors affiliated with countries belonging to two or all three world regions.

Venn diagrams were constructed for $p$-values between 0.041 and 0.049, $p$-values
between 0.051 and 0.059, and all abstracts, to illustrate the number of papers belonging to
pure disciplines/world regions and to discipline/world region intersections. Longitudinal
trends in the three pure disciplines/world regions were estimated as in "Longitudinal
trends and their comparisons between scientific disciplines and between world regions".

## Significance versus using idiosyncratic expressions in 1990 and 2013

To investigate whether the longitudinal trends of $p$-values between 0.041 and 0.049 and
between 0.051 and 0.059 differ from the corresponding trends of other manifestations of
significance, we conducted 25 additional Scopus search queries of multiple $p$-values and
$p$-value ranges (some of which were proposed in Tweets commenting on a preprint of
the present paper; e.g., *Hankins, 2014*; *Sherman, 2014*). We also conducted 23 additional
queries of textual expressions of statistical (non-)significance; some of these were proposed
by the journal editor and 18 were taken from *Hankins (2013)*. Specifically, we checked the
number of records in 1990 and 2013 for the 509 textual expressions of significance reported
by *Hankins (2013)*. From these 509, we included the 18 that yielded 10 records or more for
both 1990 and 2013. Finally, 29 queries of idiosyncratic expressions were conducted. Some
of these were related to structured reporting (e.g., "results showed that"), whereas others
were not (e.g., "room temperature") and used as controls. Table 1 provides an overview of
all search queries and corresponding syntaxes in Scopus.

For these 77 queries as well as for the 6 queries described in "Longitudinal trends and
their comparisons between scientific disciplines and between world regions" (i.e., queries
2a, b, 3a, b, and 4a, b), the ratio $(N_{2013}/T_{2013})/(N_{1990}/T_{1990})$ was calculated, where $N_{2013}$
and $N_{1990}$ are the total numbers of abstracts with terms of one of the 83 search queries
in 2013 and 1990, respectively, and $T_{2013}$ and $T_{1990}$ are the total numbers of papers with
an abstract for these two years (as derived by query 1 in "Longitudinal trends and their
comparisons between scientific disciplines and between world regions").

## Supplementary search queries

The following data were also extracted:

(1) Yearly number of papers containing $p < 0.05$ but not $p > 0.05$, $p > 0.05$ but not
$p < 0.05$, and both $p < 0.05$ and $p > 0.05$ (i.e., three search queries in total).

(2) Yearly number of papers with abstracts containing $p$-values between 0.001 and 0.099
with a 0.001 interval, that is, $p = 0.001, p = 0.002, \ldots p = 0.099$ (i.e., 99 search queries
in total).

(3) Yearly number of papers containing $p$-values between 0.041 and 0.049, $p$-values
between 0.051 and 0.059, and the total number of papers with an abstract, in three

**Table 1  All search queries and corresponding syntaxes in Scopus.**

| | | |
|---|---|---|
| 1. | All abstracts | ABS({.}) |
| 2. | <0.001 | ABS({p < 0.001} OR {p < .001} OR {p <= 0.001} OR {p <= .001} OR {p ≤ 0.001} OR {p ≤ .001}) |
| 3. | >0.001 | ABS({p > 0.001} OR {p > .001}) |
| 4. | <0.01 | ABS({p < 0.01} OR {p < .01} OR {p <= 0.01} OR {p <= .01} OR {p ≤ 0.01} OR {p ≤ .01}) |
| 5. | >0.01 | ABS({p > 0.01} OR {p > .01}) |
| 6. | <0.05 | ABS({p < 0.05} OR {p < .05} OR {p <= 0.05} OR {p <= .05} OR {p ≤ 0.05} OR {p ≤ .05}) |
| 7. | >0.05 | ABS({p > 0.05} OR {p > .05}) |
| 8. | <0.10 | ABS({p < 0.10} OR {p < .10} OR {p <= 0.10} OR {p <= .10} OR {p ≤ 0.10} OR {p ≤ .10}) |
| 9. | >0.10 | ABS({p > 0.10} OR {p > .10}) |
| 10. | =0.001 | ABS({p = 0.001} OR {p = .001}) |
| 11. | 0.002–0.005 | ABS({p = 0.002} OR {p = .002} OR {p = 0.003} OR {p = .003} OR {p = 0.004} OR {p = .004} OR {p = 0.005} OR {p = .005}) |
| 12. | 0.006–0.009 | ABS({p = 0.006} OR {p = .006} OR {p = 0.007} OR {p = .007} OR {p = 0.008} OR {p = .008} OR {p = 0.009} OR {p = .009}) |
| 13. | 0.011–0.019 | ABS({p = 0.011} OR {p = .011} OR {p = 0.012} OR {p = .012} OR {p = 0.013} OR {p = .013} OR {p = 0.014} OR {p = .014} OR {p = 0.015} OR {p = .015} OR {p = 0.016} OR {p = .016} OR {p = 0.017} OR {p = .017} OR {p = 0.018} OR {p = .018} OR {p = 0.019} OR {p = .019}) |
| 14. | 0.021–0.029 | ABS({p = 0.021} OR {p = .021} OR {p = 0.022} OR {p = .022} OR {p = 0.023} OR {p = .023} OR {p = 0.024} OR {p = .024} OR {p = 0.025} OR {p = .025} OR {p = 0.026} OR {p = .026} OR {p = 0.027} OR {p = .027} OR {p = 0.028} OR {p = .028} OR {p = 0.029} OR {p = .029}) |
| 15. | 0.031–0.039 | ABS({p = 0.031} OR {p = .031} OR {p = 0.032} OR {p = .032} OR {p = 0.033} OR {p = .033} OR {p = 0.034} OR {p = .034} OR {p = 0.035} OR {p = .035} OR {p = 0.036} OR {p = .036} OR {p = 0.037} OR {p = .037} OR {p = 0.038} OR {p = .038} OR {p = 0.039} OR {p = .039}) |
| 16. | 0.041–0.049 | ABS({p = 0.041} OR {p = .041} OR {p = 0.042} OR {p = .042} OR {p = 0.043} OR {p = .043} OR {p = 0.044} OR {p = .044} OR {p = 0.045} OR {p = .045} OR {p = 0.046} OR {p = .046} OR {p = 0.047} OR {p = .047} OR {p = 0.048} OR {p = .048} OR {p = 0.049} OR {p = .049}) |
| 17. | 0.051–0.059 | ABS({p = 0.051} OR {p = .051} OR {p = 0.052} OR {p = .052} OR {p = 0.053} OR {p = .053} OR {p = 0.054} OR {p = .054} OR {p = 0.055} OR {p = .055} OR {p = 0.056} OR {p = .056} OR {p = 0.057} OR {p = .057} OR {p = 0.058} OR {p = .058} OR {p = 0.059} OR {p = .059}) |
| 18. | 0.061–0.069 | ABS({p = 0.061} OR {p = .061} OR {p = 0.062} OR {p = .062} OR {p = 0.063} OR {p = .063} OR {p = 0.064} OR {p = .064} OR {p = 0.065} OR {p = .065} OR {p = 0.066} OR {p = .066} OR {p = 0.067} OR {p = .067} OR {p = 0.068} OR {p = .068} OR {p = 0.069} OR {p = .069}) |
| 19. | 0.071–0.079 | ABS({p = 0.071} OR {p = .071} OR {p = 0.072} OR {p = .072} OR {p = 0.073} OR {p = .073} OR {p = 0.074} OR {p = .074} OR {p = 0.075} OR {p = .075} OR {p = 0.076} OR {p = .076} OR {p = 0.077} OR {p = .077} OR {p = 0.078} OR {p = .078} OR {p = 0.079} OR {p = .079}) |
| 20. | 0.081–0.089 | ABS({p = 0.081} OR {p = .081} OR {p = 0.082} OR {p = .082} OR {p = 0.083} OR {p = .083} OR {p = 0.084} OR {p = .084} OR {p = 0.085} OR {p = .085} OR {p = 0.086} OR {p = .086} OR {p = 0.087} OR {p = .087} OR {p = 0.088} OR {p = .088} OR {p = 0.089} OR {p = .089}) |
| 21. | 0.091–0.099 | ABS({p = 0.091} OR {p = .091} OR {p = 0.092} OR {p = .092} OR {p = 0.093} OR {p = .093} OR {p = 0.094} OR {p = .094} OR {p = 0.095} OR {p = .095} OR {p = 0.096} OR {p = .096} OR {p = 0.097} OR {p = .097} OR {p = 0.098} OR {p = .098} OR {p = 0.099} OR {p = .099}) |
| 22. | 0.01 | ABS({p = 0.010} OR {p = .010} OR {p = 0.01} OR {p = .01}) |
| 23. | 0.02 | ABS({p = 0.020} OR {p = .020} OR {p = 0.02} OR {p = .02}) |
| 24. | 0.03 | ABS({p = 0.030} OR {p = .030} OR {p = 0.03} OR {p = .03}) |
| 25. | 0.04 | ABS({p = 0.040} OR {p = .040} OR {p = 0.04} OR {p = .04}) |
| 26. | 0.05 | ABS({p = 0.050} OR {p = .050} OR {p = 0.05} OR {p = .05}) |

Table 1 (*continued*)

| | | |
|---|---|---|
| 27. | 0.06 | ABS({p = 0.060} OR {p = .060} OR {p = 0.06} OR {p = .06}) |
| 28. | 0.07 | ABS({p = 0.070} OR {p = .070} OR {p = 0.07} OR {p = .07}) |
| 29. | 0.08 | ABS({p = 0.080} OR {p = .080} OR {p = 0.08} OR {p = .08}) |
| 30. | 0.09 | ABS({p = 0.090} OR {p = .090} OR {p = 0.09} OR {p = .09}) |
| 31. | p = NS or p = N.S. | ABS({p = NS} OR {p = N.S.}) |
| 32. | "significant difference(s)" | ABS(({significant difference} OR {significant differences} OR {significantly different} OR {differed significantly}) AND NOT ({no significant difference} OR {no significant differences} OR {no statistically significant difference} OR {no statistically significant differences} OR {not significantly different} OR {did not differ significantly})) |
| 33. | "no significant difference(s)" | ABS({no significant difference} OR {no significant differences} OR {no statistically significant difference} OR {no statistically significant differences} OR {not significantly different} OR {did not differ significantly}) |
| 34. | "significant effect(s)" | ABS(({significant effect} OR {significant effects}) AND NOT ({no significant effect} OR {no significant effects} OR {no statistically significant effect} OR {no statistically significant effects} OR {not a significant effect} OR {not a statistically significant effect})) |
| 35. | "no significant effect(s)" | ABS({no significant effect} OR {no significant effects} OR {no statistically significant effect} OR {no statistically significant effects} OR {not a significant effect} OR {not a statistically significant effect}) |
| 36. | "supports the hypothesis" | ABS(({supports the hypothesis} OR {support the hypothesis} OR {supports our hypothesis} OR {support our hypothesis} ) AND NOT ({does not support the hypothesis} OR {do not support the hypothesis} OR {does not support our hypothesis} OR {do not support our hypothesis})) |
| 37. | "does not support the hypothesis" | ABS({does not support the hypothesis} OR {do not support the hypothesis} OR {does not support our hypothesis} OR {do not support our hypothesis}) |
| 38. | "significantly higher/more" | ABS({significantly higher} OR {significantly more}) |
| 39. | "significantly lower/less" | ABS({significantly lower} OR {significantly less}) |
| 40. | "marginally significant" | ABS("marginally significant") |
| 41. | "important finding" | ABS("important finding" OR "important findings") |
| 42. | "pH 7" | ABS("ph 7") |
| 43. | "mass of" | ABS("mass of") |
| 44. | "room temperature" | ABS("room temperature") |
| 45. | "melting point" | ABS("melting point") |
| 46. | "field of view" | ABS("field of view") |
| 47. | "the properties of" | ABS("the properties of") |
| 48. | "the aim of" | ABS("the aim of") |
| 49. | "our aim" | ABS("our aim") |
| 50. | "results showed that" | ABS("results showed that") |
| 51. | "in conclusion" | ABS("in conclusion") |
| 52. | "longitudinal study" | ABS("longitudinal study") |
| 53. | "in other words" | ABS("in other words") |
| 54. | "on the other hand" | ABS("on the other hand") |
| 55. | "a novel" | ABS("a novel") |
| 56. | "a new" | ABS("a new") |
| 57. | "was/were measured" | ABS("was measured" OR "were measured") |
| 58. | "we measured" | ABS("we measured") |
| 59. | "paradigm shift" | ABS("paradigm shift") |
| 60. | data | ABS(data) |

Table 1 (*continued*)

| 61. | information | ABS(information) |
|---|---|---|
| 62. | experiment | ABS(experiment) |
| 63. | important | ABS(important) |
| 64. | interesting | ABS(interesting) |
| 65. | neutral | ABS(neutral) |
| 66. | positive | ABS(positive) |
| 67. | negative | ABS(negative) |
| 68. | "highly significant" | ABS("highly significant" AND NOT "not highly significant") |
| 69. | "trend toward" | ABS("trend toward") |
| 70. | "an increasing trend" | ABS("an increasing trend") |
| 71. | "a decreasing trend" | ABS("a decreasing trend") |
| 72. | "potentially significant" | ABS("potentially significant") |
| 73. | "a nonsignificant trend" | ABS("a nonsignificant trend" OR "a non significant trend") |
| 74. | "a significant trend" | ABS("a significant trend") |
| 75. | "quite significant" | ABS("quite significant") |
| 76. | "a clear trend" | ABS("a clear trend") |
| 77. | "a positive trend" | ABS("a positive trend") |
| 78. | "a strong trend" | ABS("a strong trend") |
| 79. | "significant tendency" | ABS("significant tendency") |
| 80. | "a little significant" | ABS("a little significant") |
| 81. | "not insignificant" | ABS("not insignificant") |
| 82. | "possible significance" | ABS("possible significance") |
| 83. | "failed to reach statistical significance" | ABS("failed to reach statistical significance") |
| 84. | "likely to be significant" | ABS("likely to be significant") |

research fields of biological sciences, proposed by one of the reviewers: psychiatry, surgery, and cardiovascular medicine. This was done by restricting our searches to the terms psychiatr*, surg*, and cardio* OR heart, respectively, in the journal name (i.e., 3 search terms x 3 research fields = 9 search queries).

## General methodological considerations

Each abstract was counted only once per search query, independent of whether it included one or more manifestations of significance. All data were extracted between 15 and 30 November 2014. All searches of numerical $p$-values were conducted both for strings with and without zero before the decimal separator (e.g., 0.05 and .05). For all search queries containing "<" or ">" we also included variants with "<=," "=<," "≤" or ">=," "=>," "≥." Searches in Scopus are not case sensitive, that is, a search query in lower cases (e.g., {$p = 0.001$}) also retrieves records with the same query in upper cases (e.g., P = 0.001). Spaces around mathematical operators are neglected by Scopus; that is, queries with and without spaces yield the same results. All analyses were conducted for papers published in the period 1990–2013. The employed MATLAB (version R2011b) script, the CSV files with the raw data as exported from Scopus, and an Excel file with all search syntaxes and the names of the corresponding CSV files are provided as Supplemental Information

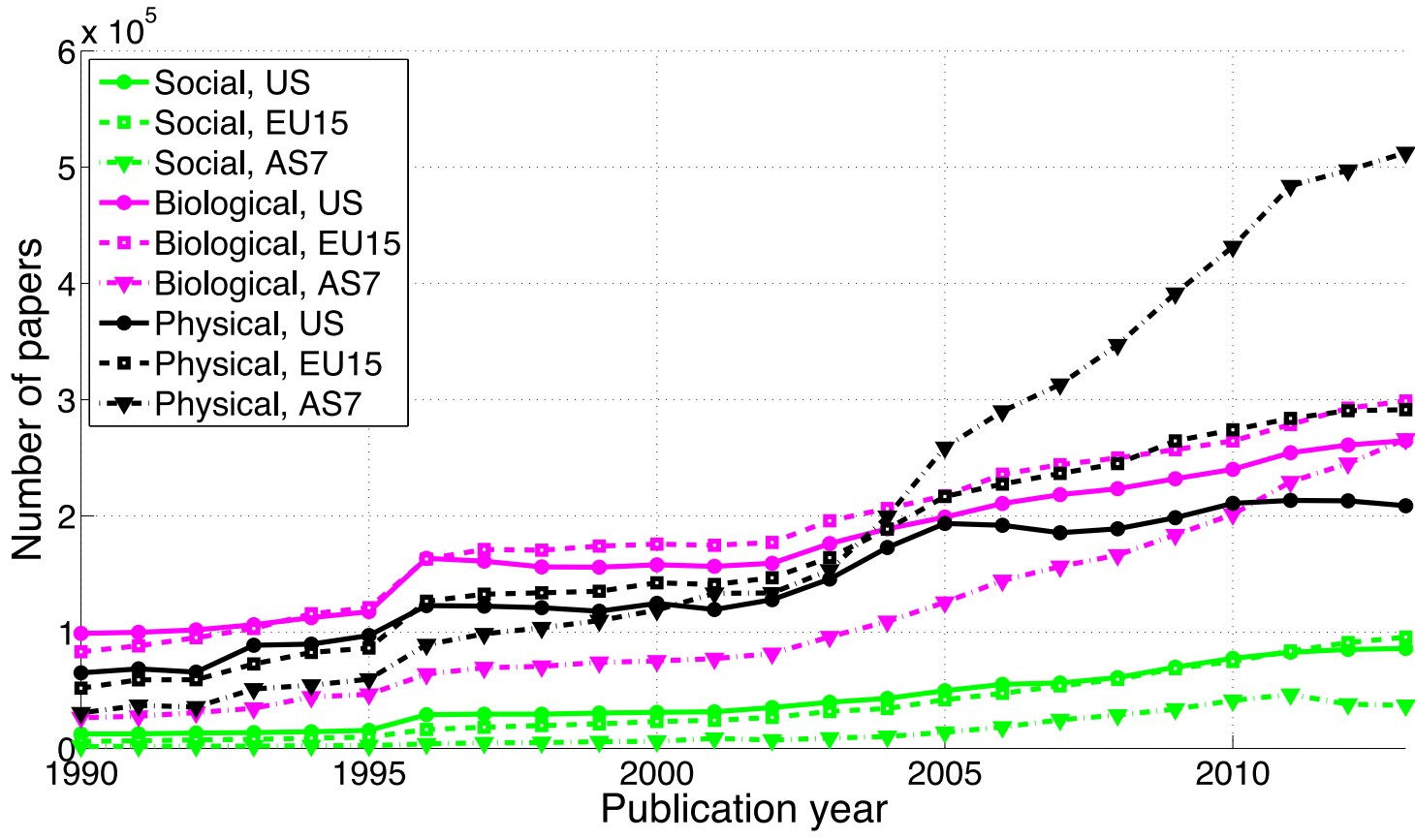

**Figure 2** Number of papers per publication year, for three scientific disciplines and three world regions.

## RESULTS

### Total number of papers

According to our searches, Scopus contained a total of 30,677,779 papers with an abstract published between 1990 and 2013. Of these, 3,061,170 papers belonged to the social sciences, 14,412,460 papers belonged to the biological sciences, and 15,364,142 papers belonged to the physical sciences. The sum of the number of papers in the three disciplines was greater than the total number of papers, because some papers were classified into multiple disciplines. The US were found in the affiliations of 8,073,346 papers, EU15 in 8,790,965 papers, and AS7 in 7,413,236 papers.

The number of papers has increased over the years in all three scientific disciplines (Fig. 2). Over the period 1990–2013, papers in the social sciences represented 12.4% (i.e., 100%*1,003,555/8,073,346) of all papers from the US, 10.0% of all papers from the EU15, and only 4.8% of all papers from the AS7. Papers in the biological sciences represented 52.2% of all papers from the US, 51.8% of all papers from the EU15, and 35.7% of all papers from the AS7. Papers in the physical sciences represented 42.8% of all papers from the US, 46.1% of all papers from the EU15, and 66.6% of all papers from the AS7. In other words, AS7 has been publishing considerably less than the US and EU15

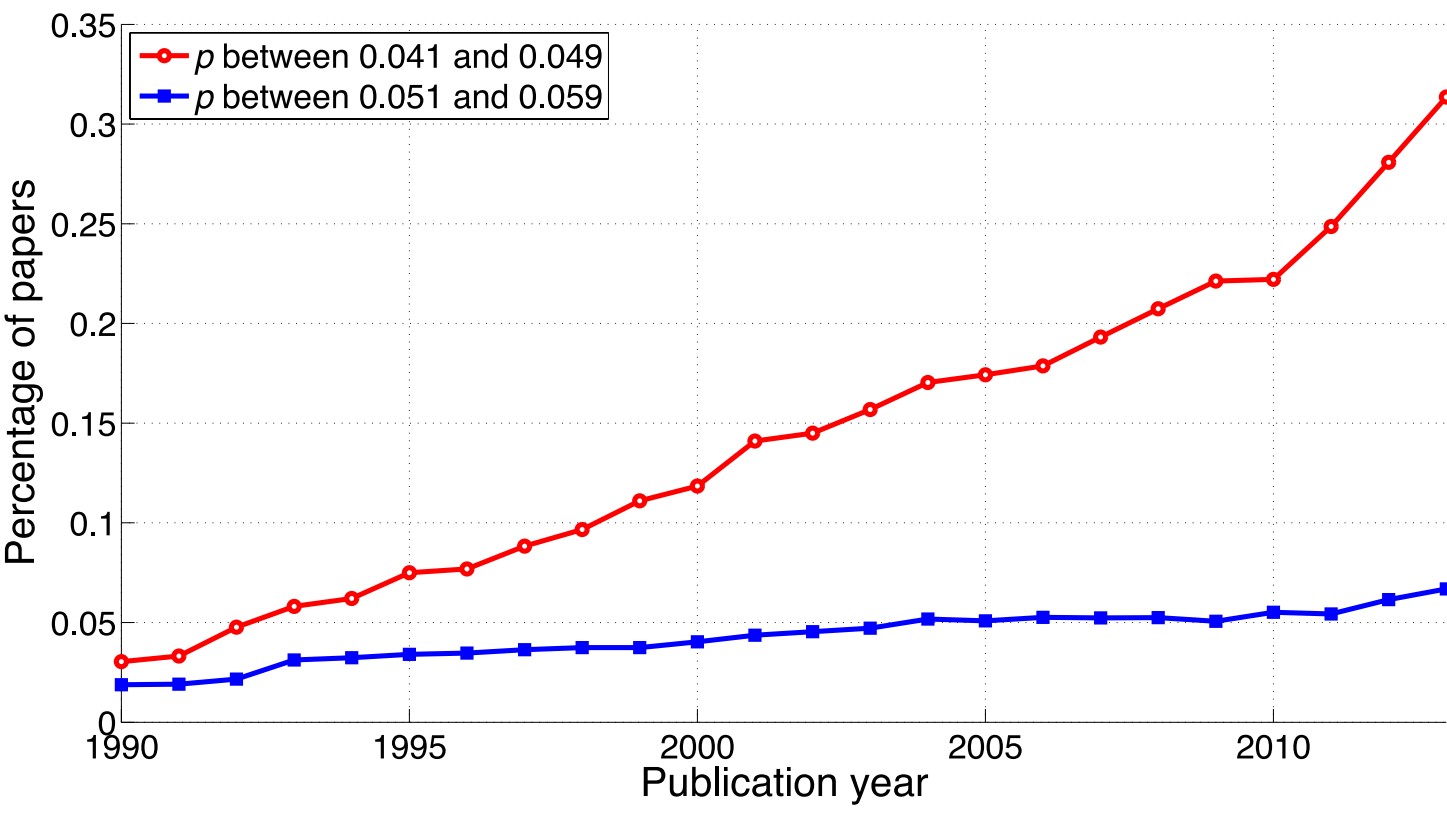

**Figure 3** Percentage of papers reporting a *p*-value between 0.041 and 0.049 and percentage of papers reporting a *p*-value between 0.051 and 0.059 per publication year.

in the social sciences and considerably more in the physical sciences. AS7's production of physical sciences papers was half the production of physical sciences papers in the US in 1990 (physical sciences papers AS7/physical sciences papers US = 30,558/65,045 = 0.47) and more than double the production of physical sciences papers in the US in 2013 (physical sciences papers AS7/physical sciences papers US = 512,513/208,698 = 2.46).

### Longitudinal trends

#### *p-values between 0.041 and 0.049 versus p-values between 0.051 and 0.059*

Both the *p*-values between 0.041 and 0.049 and the *p*-values between 0.051 and 0.059 have increased over time (Fig. 3). In 1990, 0.019% of papers (106 out of 563,023 papers) reported a *p*-value between 0.051 and 0.059. This increased 3.6-fold to 0.067% (1,549 out of 2,317,062 papers) in 2013. Positive results increased 10.3-fold in the same period: from 0.030% (171 out of 563,023 papers) in 1990 to 0.314% (7,266 out of 2,317,062 papers) in 2013. In other words, the ratio of significant to non-significant results increased from 1.6 (i.e., 171/106 papers) in 1990 to 4.7 (i.e., 7,266/1,549 papers) in 2013 (Fig. 4).

#### *"Significant difference" versus "no significant difference"*

The reporting of "significant difference" has increased over time (Fig. 5; 2013-to-1990 ratio of the percentage of papers = 1.5), whereas the reporting of "no significant difference" has

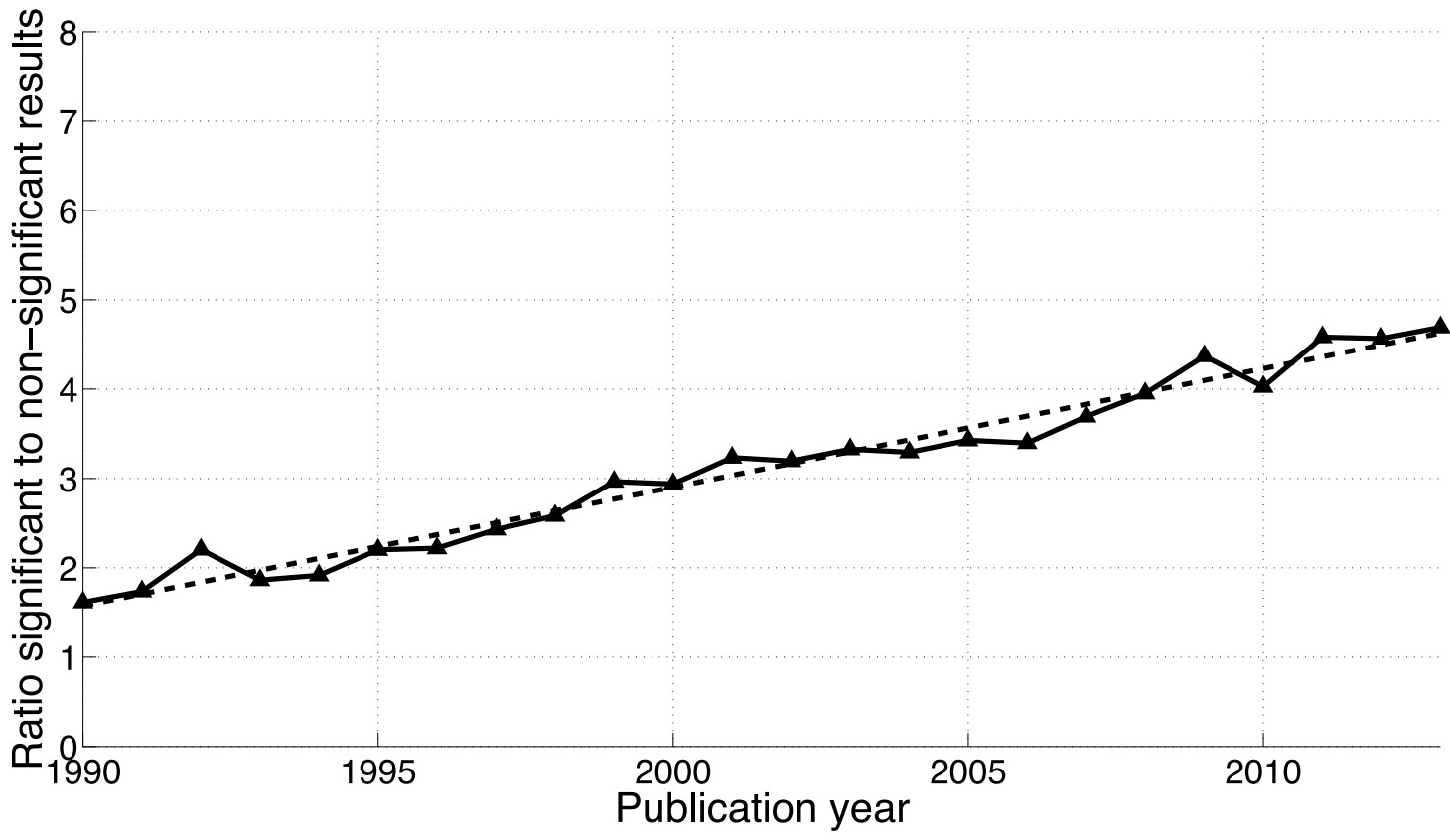

**Figure 4 Ratio of *p*-values between 0.041 and 0.049 to *p*-values between 0.051 and 0.059 per publication year.** The dashed line represents the result of a simple linear regression analysis.

remained about the same (ratio = 1.1). Compared to the findings on *p*-values described above, the ratio of "significant difference" to "no significant difference" was lower and increased less rapidly (Fig. 4 versus Fig. 6). Specifically, the ratio of "significant difference" to "no significant difference" increased from 0.60 (i.e., 4,405/7,308 papers) in 1990 to 0.85 (i.e., 27,816/32,821 papers) in 2013. Averaged over the period 1990–2013, the phrase "no significant difference" was 1.4 times more prevalent than the phrase "significant difference."

### *p* < 0.05 *versus p* > 0.05

Both $p < 0.05$ and $p > 0.05$ have increased over the years, in line with the findings described in the previous two sections. The increase of reporting of $p > 0.05$ however, was steeper (2013-to-1990 ratio of the percentage of papers = 4.8) than that of $p < 0.05$ (2013-to-1990 ratio of the percentage of papers = 1.4; Fig. 7), opposite to the trends of *p*-values and of "significant difference" versus "no significant difference." The ratio of $p < 0.05$ to $p > 0.05$ *decreased* from 19.7 (i.e., 7,193/366 papers) in 1990 to 5.6 (i.e., 40,904/7,255 papers) in 2013 (Fig. 8).

In a supplementary analysis we searched for and compared abstracts containing $p < 0.05$ but not $p > 0.05$, abstracts containing $p > 0.05$ but not $p < 0.05$, and abstracts

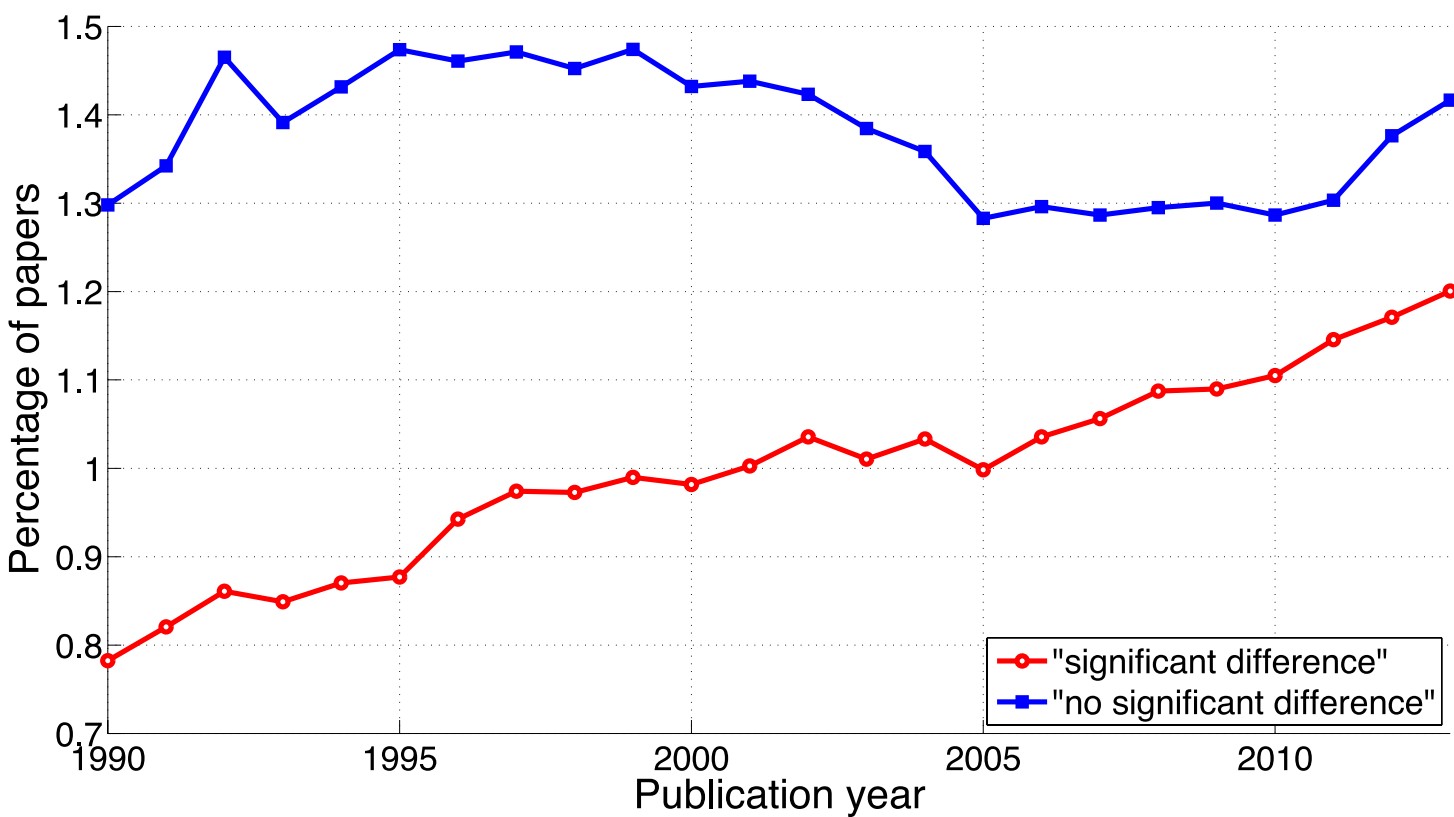

**Figure 5** Percentage of papers reporting "significant difference" and percentage of papers reporting "no significant difference" per publication year.

containing both $p < 0.05$ and $p > 0.05$. We found that the percentage of abstracts reporting only $p < 0.05$ was considerably larger (1.6% averaged over 1990–2013) than the percentage of abstracts reporting only $p > 0.05$ (0.12%) and the percentage of abstracts reporting both $p < 0.05$ and $p > 0.05$ (0.07%; see Fig. S1). The increase of reporting $p > 0.05$ only was steeper (2013-to-1990 ratio of the percentage of papers = 2.4) than that of $p < 0.05$ only (2013-to-1990 ratio of the percentage of papers = 1.3), in line with the results of reporting $p < 0.05$ versus $p > 0.05$.

### Comparison of longitudinal trends between scientific disciplines

#### *p-values between 0.041 and 0.049 versus p-values between 0.051 and 0.059*

A comparison between disciplines shows that the use of $p$-values between 0.041 and 0.049 and between 0.051 and 0.059 was rare in the social sciences and even rarer in the physical sciences (Fig. 9). The $p$-values between 0.041 and 0.049 over 1990–2013 were 65.7 times more prevalent in the biological sciences than in the physical sciences and 8.6 times more prevalent in the biological sciences than in the social sciences. Over 1990–2013, $p$-values between 0.051 and 0.059 were respectively 47.3 and 8.7 times more prevalent in the biological sciences than in the physical and in the social sciences (Fig. 9). The

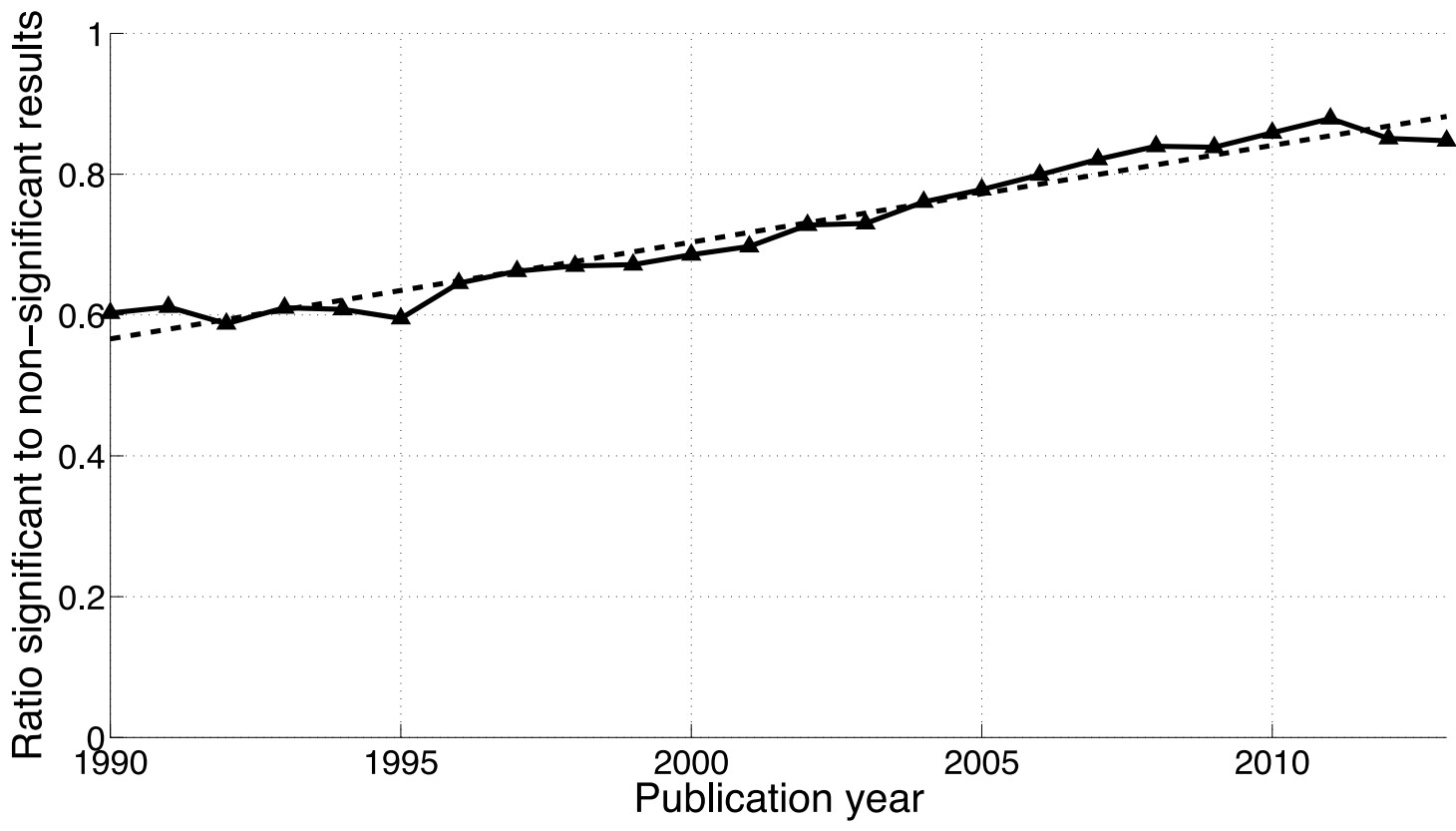

**Figure 6 Ratio of "significant difference" to "no significant difference" per publication year.** The dashed line represents the result of a simple linear regression analysis.

regression slopes of the ratios of significant to non-significant results were similar for the three disciplines (Figs. 10 and 16).

### "Significant difference" versus "no significant difference"

Figure 11 shows that the biological sciences were more likely to use the phrases "significant difference" and "no significant difference" than the social sciences. These expressions were rare in the physical sciences, confirming the results for $p$-values in Fig. 9. Physical sciences exhibited the highest overall ratio of significant to non-significant results (Fig. 12), whereas the growth rate of the ratio of significant to non-significant results was higher in the social sciences than in the biological sciences, the growth rate of which in turn was higher than that in the physical sciences (Fig. 16).

### $p < 0.05$ versus $p > 0.05$

The use of $p < 0.05$ was rare in the social and physical sciences (Fig. 13). The ratio of significant-over-non-significant results was highest for the social sciences and decreased more rapidly for the social sciences than for the physical sciences (Fig. 14).

### Pure disciplines

The Venn diagrams in Fig. 15 show the number of papers belonging to pure disciplines, as well as the number of papers classified into two or all three disciplines. It can be seen

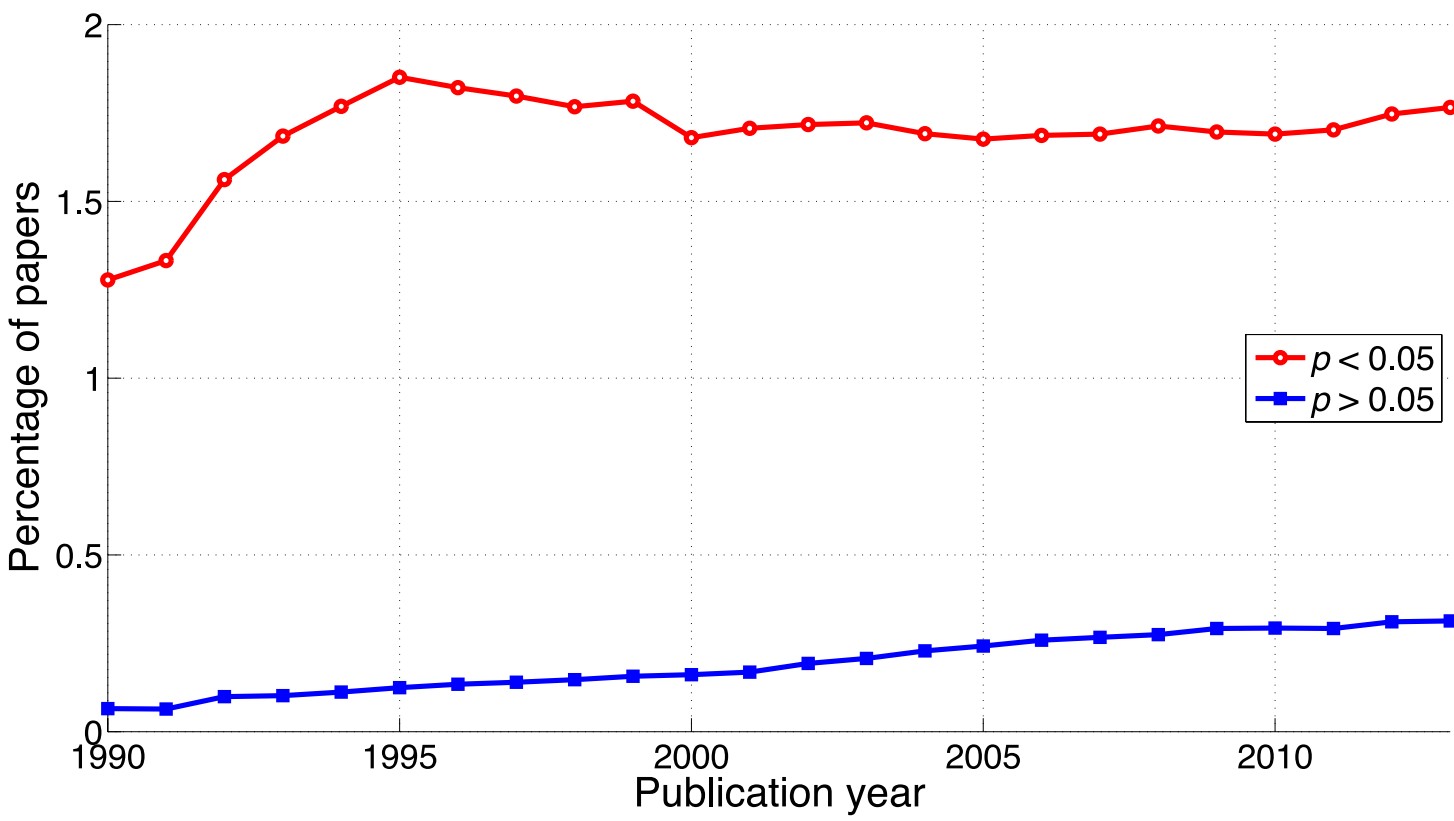

**Figure 7** Percentage of papers reporting $p < 0.05$ and percentage of papers reporting $p > 0.05$ per publication year.

that the greatest numbers of papers reporting $p$-values between 0.041 and 0.049 or between 0.051 and 0.059 belonged to the biological sciences; these $p$-values were rare in the pure social sciences, and even rarer in the pure physical sciences. The Venn diagrams further show that *if* a $p$-value were used in the physical sciences, it often occurred in papers with biological affinity (i.e., 670 out of 798 the physical sciences papers with $p$-values between 0.041 and 0.049 and 239 out the 286 physical sciences papers with $p$-values between 0.051 and 0.059 were cross-classified into the biological sciences). A large overlap also occurred between the social sciences and the biological sciences.

Figure 16 shows the slope coefficients calculated using a simple linear regression analysis for the ratios of significant to non-significant results and the percentages of significant results. These slopes were calculated for $p$-values between 0.041 and 0.049 versus $p$-values between 0.051 and 0.059, "significant difference" versus "no significant difference," and $p < 0.05$ versus $p > 0.05$, for the three disciplines, both for cross-classified papers and pure disciplines. It can be seen that the results for pure disciplines were similar to the results for the cross-classified papers described in "$p$-values between 0.041 and 0.049 versus $p$-values between 0.051 and 0.059"–"$p < 0.05$ versus $p > 0.05$." The 95% confidence intervals were wider and the trends were more erratic for the analysis of pure disciplines (and for the social and physical sciences in particular) as compared to the analysis of cross-classified

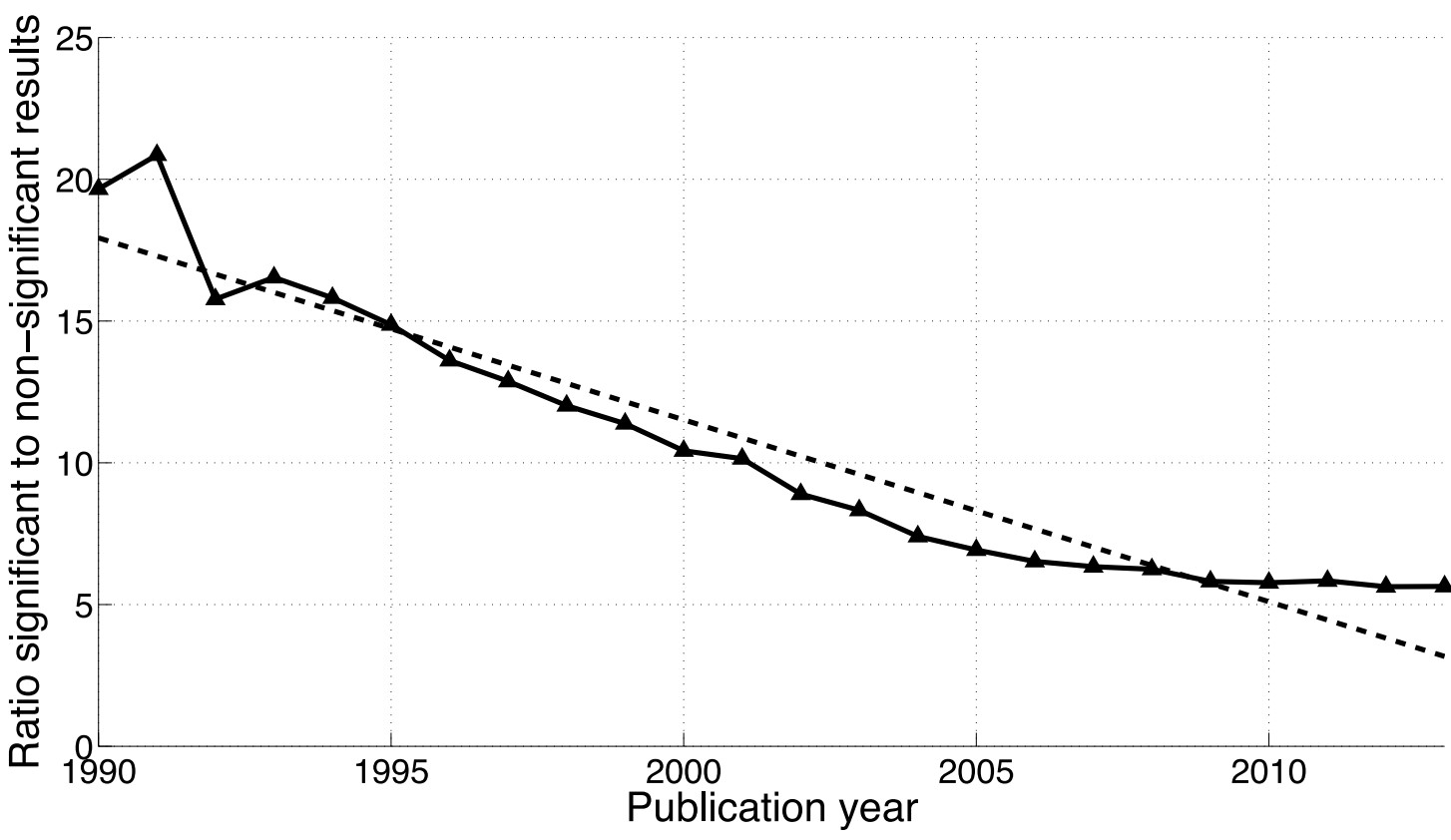

**Figure 8 Ratio of $p < 0.05$ to $p > 0.05$ per publication year.** The dashed line represents the result of a simple linear regression analysis.

papers, because the sample sizes were smaller in the former case (see also left and middle Venn diagrams in Fig. 15).

## Comparison of longitudinal trends between world regions

### p-values between 0.041 and 0.049 versus p-values between 0.051 and 0.059

For the three world regions, both $p$-values between 0.041 and 0.049 and $p$-values between 0.051 and 0.059 have increased over time (see Fig. S2). The growth of the reporting significant results as compared to the reporting of non-significant results was greater in AS7 than in the US and EU15 (Fig. 17).

### "Significant difference" versus "no significant difference"

The use of "significant difference" increased while the use of "no significance difference" decreased for all three world regions (Fig. S3). While AS7 exhibited the highest ratio of significant to non-significant results in terms of $p$-value reporting (Fig. 17), this region showed the lowest ratio of "significant difference" to "no significant difference" (Fig. 18).

### p < 0.05 versus p > 0.05

Reporting $p < 0.05$ versus $p > 0.05$ differed from both the reporting of $p$-values (0.041–0.049 versus 0.051–0.059) and "significant difference" versus "no significant difference," with a *decrease* of significant ($p < 0.05$) results for US and EU15 over time, an

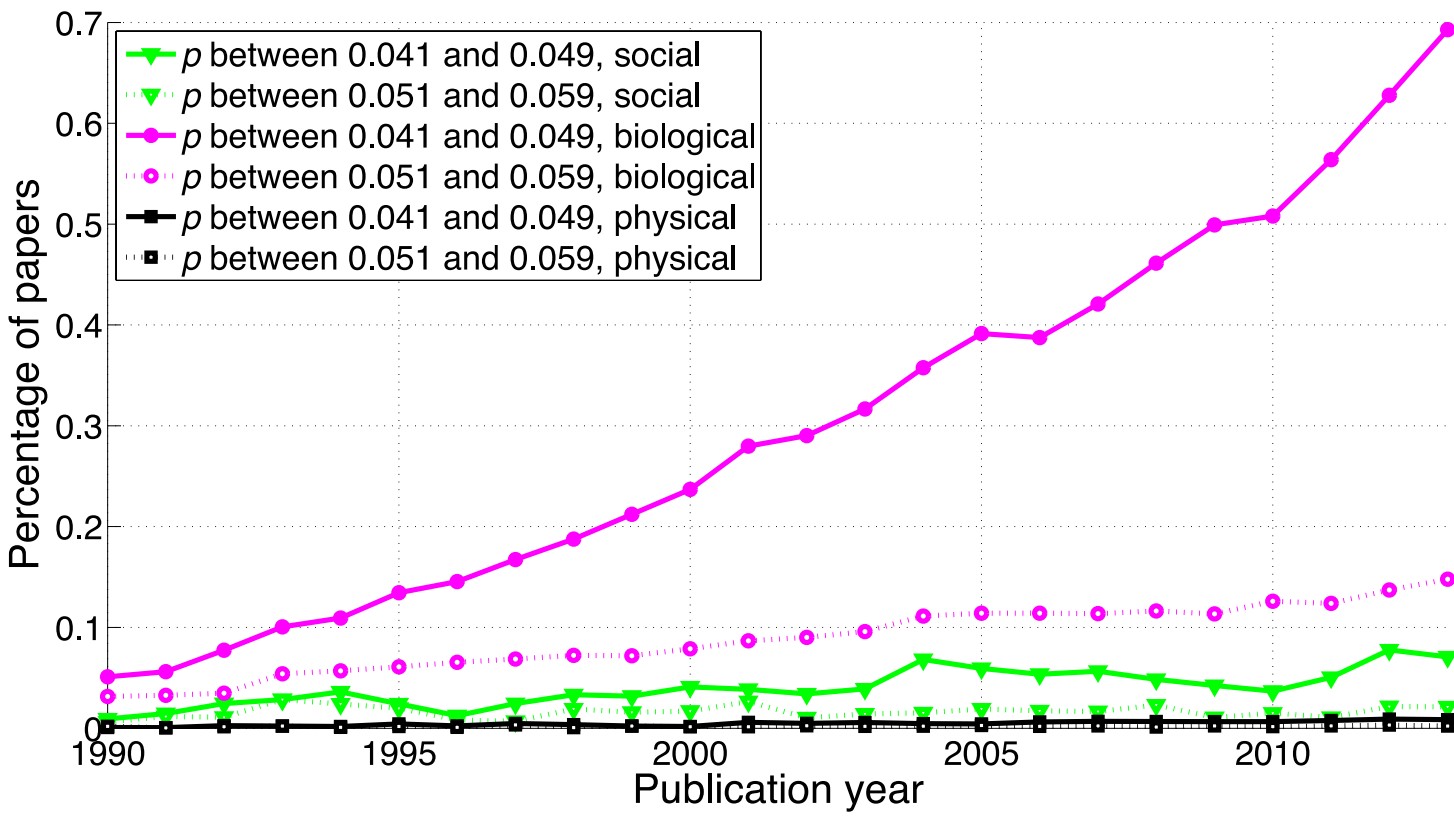

**Figure 9** Percentage of papers reporting a *p*-value between 0.041 and 0.049 and percentage of papers reporting a *p*-value between 0.051 and 0.059 per publication year, for three scientific disciplines.

increase of both significant and non-significant results for AS7 (Fig. S4), and a decreasing ratio of significant to non-significant results for all three world regions (Fig. 19).

### Pure world regions

Figure 20 shows the number of papers all authors of which were affiliated with only one world region (i.e., pure world regions), as well as the number of papers with authors from different world regions. It can be seen that between 82% and 91% of all papers reporting *p*-values between 0.041 and 0.049 and between 0.051 and 0.059 were affiliated solely with one world region.

Figure 21 shows the slope coefficients calculated using a simple linear regression for the ratios of significant to non-significant results and the percentages of significant results. These slopes were calculated for *p*-values between 0.041 and 0.049 versus *p*-values between 0.051 and 0.059, "significant difference" versus "no significant difference," and $p < 0.05$ versus $p > 0.05$, for the three world regions, both for cross-classified papers and pure world regions. The results for pure world regions were comparable to the results for the cross-classified papers described in "*p*-values between 0.041 and 0.049 versus *p*-values between 0.051 and 0.059" −"$p < 0.05$ versus $p > 0.05$."

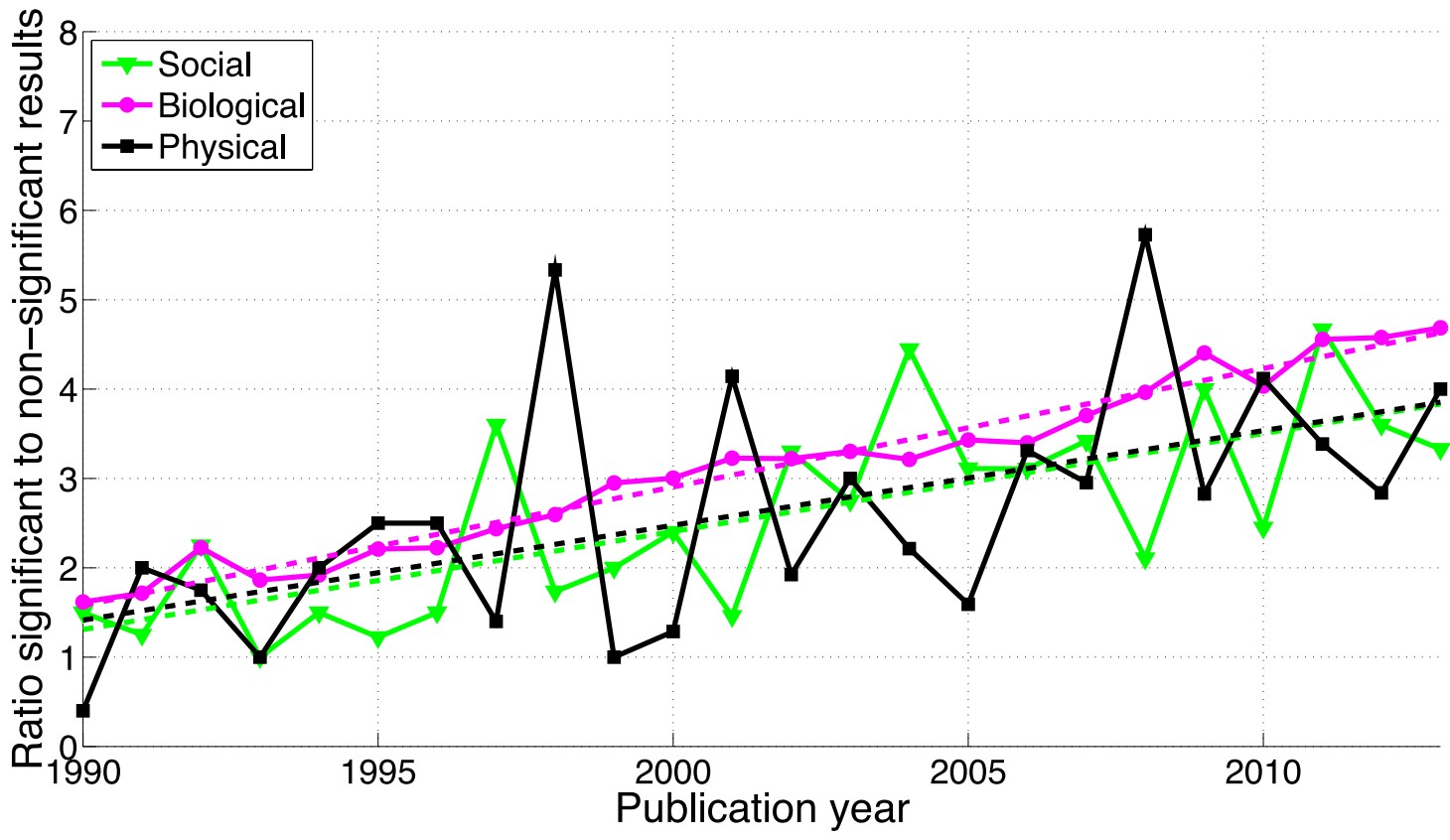

**Figure 10** **Ratio of *p*-values between 0.041 to 0.049 to *p*-values between 0.051 and 0.059 per publication year, for three scientific disciplines.** The dashed lines represent the results of a simple linear regression analysis.

### Significance versus using idiosyncratic expressions in 1990 and 2013

Figure 22 shows the 2013-to-1990 ratio of the percentage of papers for 83 search queries. It can be seen that *p*-value reporting has increased more rapidly than the reporting of the terms "significant difference" and "no significant difference." Among all ranges of three-digit *p*-values below 0.1 tested, the 2013-to-1990 ratio was highest for *p*-values between 0.041 and 0.049, and lowest for *p*-values between 0.051 and 0.059. The *p*-values with two decimal numbers showed the same pattern, with the largest increases for $p = 0.04$ and the smallest increases for $p = 0.05$. The use of some idiosyncratic expressions related to structured reporting, such as "our aim," "the aim of," and "results showed that," has increased over the years (2013-to-1990 ratio = 8.3, 5.5, & 5.0, respectively). Note that in terms of prevalence, the smaller the *p*-value the more frequently it was reported (see numbers at right of the bars at Fig. 22). This finding is confirmed by a supplementary analysis (Fig. 23), in which we investigated the prevalence of *p*-values between 0.001 and 0.099 with finer (0.001) intervals than the intervals in Fig. 22.

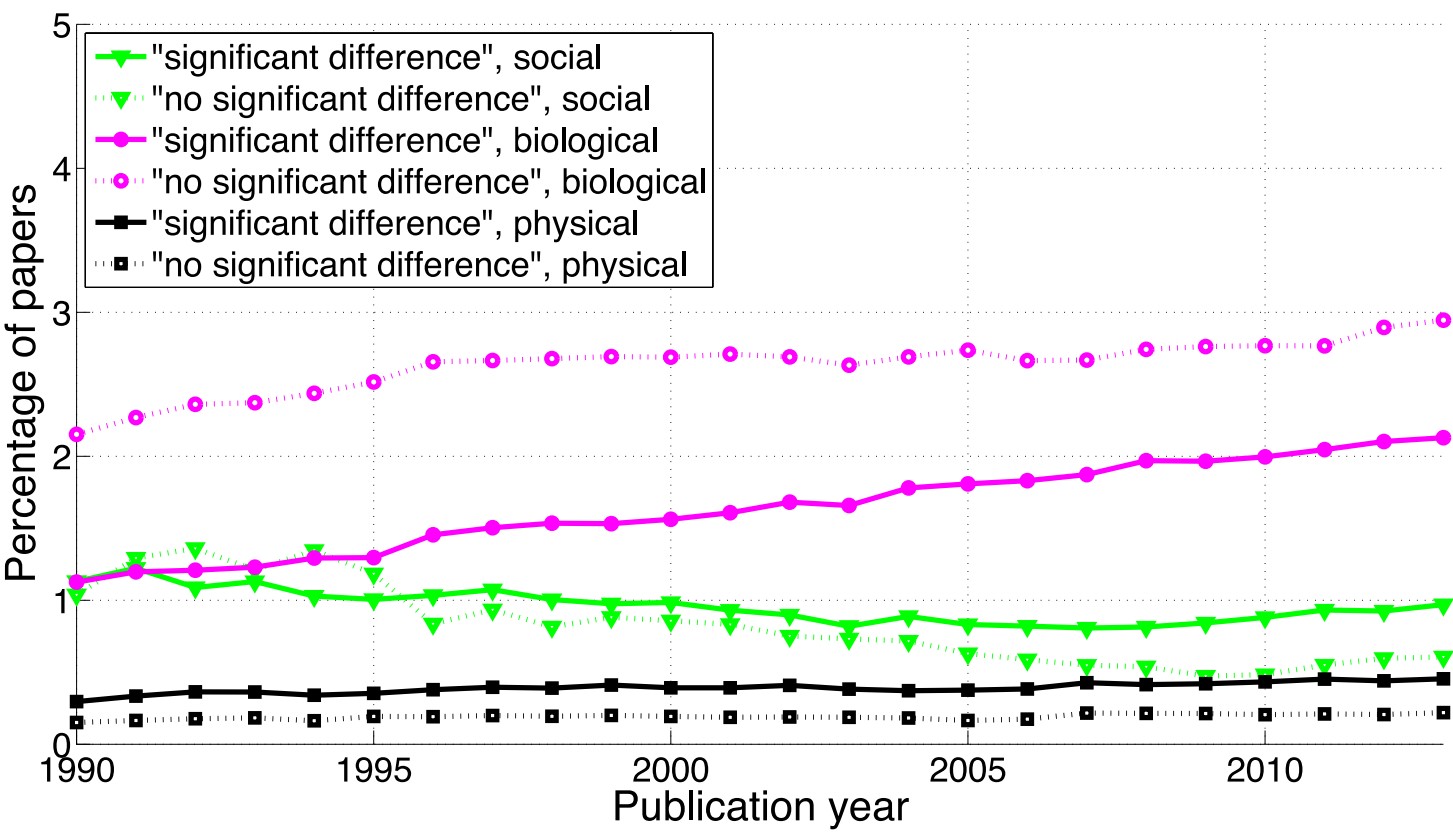

**Figure 11 Percentage of papers reporting "significant difference" and percentage of papers reporting "no significant difference" per publication year, for three scientific disciplines.**

## DISCUSSION

### Longitudinal trends of (non-)significance reporting

We investigated longitudinal trends between 1990 and 2013 for $p$-values between 0.041 and 0.049 versus $p$-values between 0.051 and 0.059, for "significant difference" versus "no significant difference," and for $p < 0.05$ versus $p > 0.05$. The longitudinal trends were compared between disciplines and between world regions. A comparison of the reporting of a variety of $p$-values, $p$-value ranges, textual reporting of significance, and idiosyncratic expressions (a total of 83 search queries) in 1990 versus 2013 was conducted as well.

The percentage of papers with $p$-values between 0.051 and 0.059 in the abstract has increased by a factor of 3.6 between 1990 and 2013, which indicates that negative results are *not* disappearing. Also striking is the 10.3-fold increase of $p$-values between 0.041 and 0.049 over the same time period.

For three-digit $p$-values below 0.1, the smaller the $p$-value the more frequently it was reported (Figs. 22 and 23), in line with the right skewed distribution of $p$-values that is to be expected when the null hypothesis is false (*Simonsohn, Nelson & Simmons, 2014*). In other words, we did not find a peak of $p$-values just below 0.05. We did find, however, a longitudinal change in the $p$-value distribution, with $p$-values between 0.041 and 0.049 showing the most rapid increase over the years. Among the three-digit $p$-value ranges we

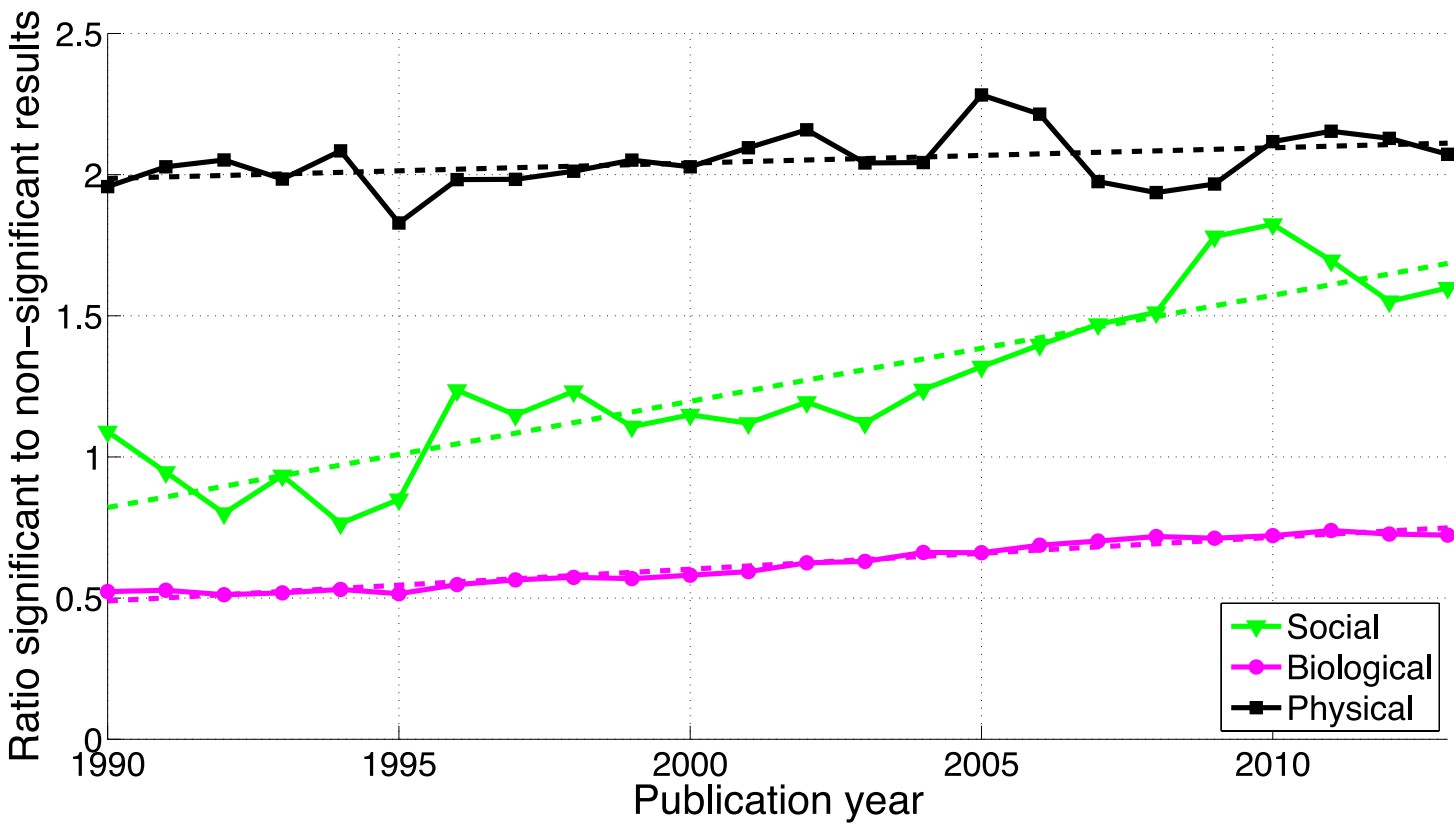

**Figure 12** Ratio of "significant difference" to "no significant difference" per publication year, for three scientific disciplines. The dashed lines represent the results of a simple linear regression analysis.

searched for, the 2013-to-1990 ratio of the percentage of papers was highest for $p$-values between 0.041 and 0.049, and lowest for $p$-values between 0.051 and 0.059. The two-digit $p$-values showed the same pattern, with $p = 0.04$ exhibiting the largest increase and $p = 0.05$ the smallest increase from 1990 to 2013.

Reporting of "significant difference" versus "no significant difference" displayed a more modest increase than $p$-value reporting, resembling *Pautasso*'s (*2010*) findings. The use of $p > 0.05$ has increased more steeply than $p < 0.05$, meaning that the corresponding regression slope is negative (Figs. 16 and 21). We conclude that the prevalence and growth rates of significance and non-significance reporting strongly depend on the search term.

The use of all $p$-values has increased rapidly since 1990 (between 2 and 10 times; Fig. 22), suggesting that null hypothesis significance testing has become more widely used over the years, despite widespread criticism against the use of $p$-values (see e.g., *Wagenmakers, 2007*). The idiosyncratic expression "paradigm shift" was used 15.2 times more often in 2013 than in 1990 (cf. *Atkin, 2002*, who reported that the expression was found in 4 titles published in 1990 versus 30 in 2000 and 22 in 2001).

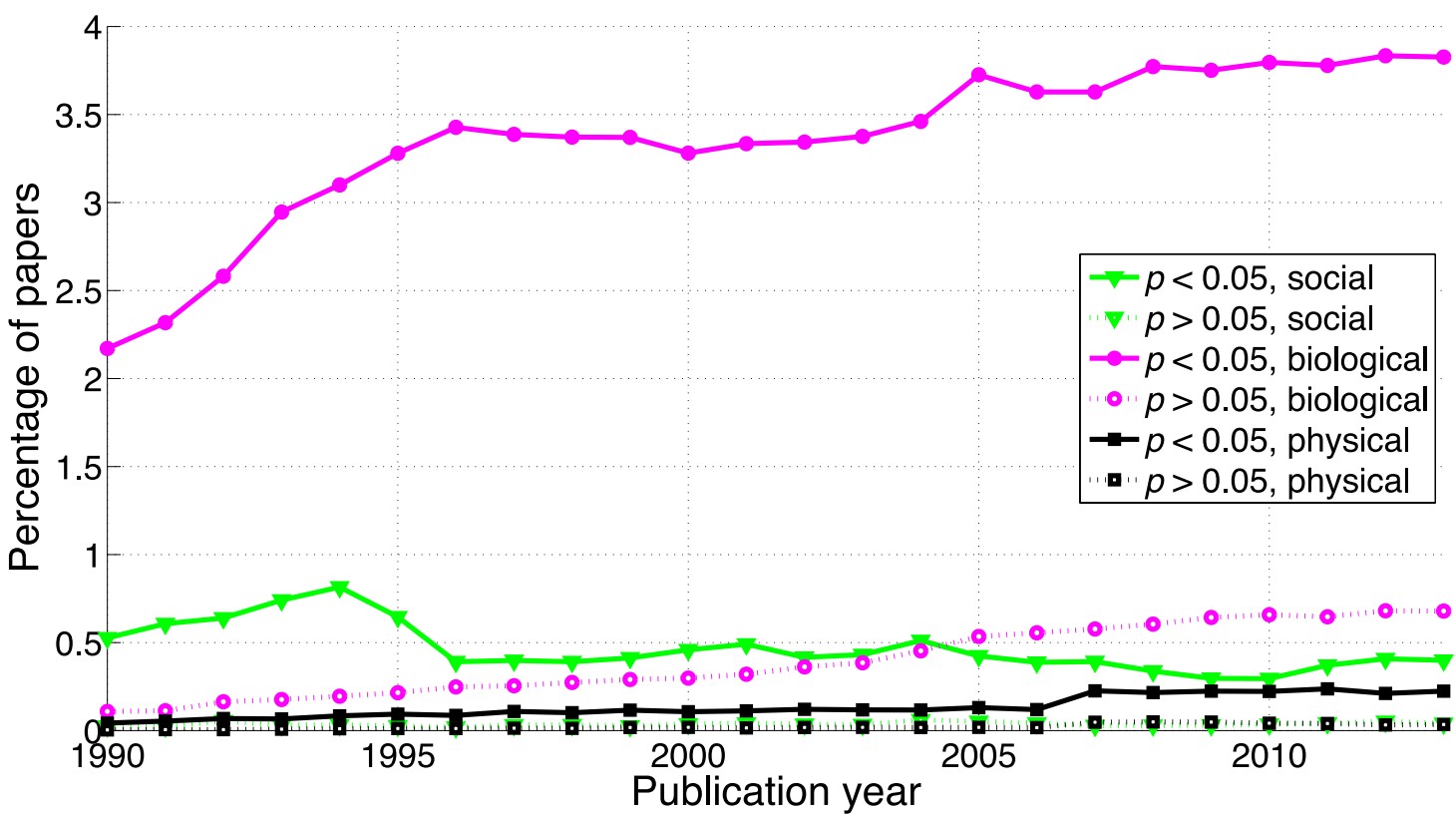

**Figure 13** **Percentage of papers reporting $p < 0.05$ and percentage of papers reporting $p > 0.05$ per publication year, for three scientific disciplines.**

## Comparisons of longitudinal trends between scientific disciplines

We found no support for the hierarchy of sciences as discussed by *Fanelli (2010)*. In our analysis, the differences in the growth rates of the positive to negative results ratios between the three scientific disciplines were inconsistent and dependent on the search term. For example, for the ratio of "significant difference" to "no significant difference," social sciences exhibited a significantly steeper increase than the physical sciences. Contrarily, for $p < 0.05$ versus $p > 0.05$, social sciences exhibited a significantly steeper *decrease* than the physical sciences (Fig. 16).

The more salient finding of our analysis is the enormous differences in reporting practices between disciplines, with the use of $p$-values between 0.041 and 0.049 being 65.7 times more prevalent in the biological sciences than in the physical sciences. The prevalence of $p$-values between 0.041 and 0.049 in the biological sciences is even larger when considering that many physical sciences papers reporting $p$-values had medical affinity (e.g., a new ultrasound techniques or a new radiology method being tested in a medical setting). Indeed, 84% of the physical sciences papers containing $p$-values between 0.041 and 0.049 were also classified into biological sciences by Scopus (see the left Venn diagram in Fig. 15). The $p$-values between 0.041 and 0.049 were an astonishing 387 times more prevalent in the pure biological sciences than in the pure physical sciences.

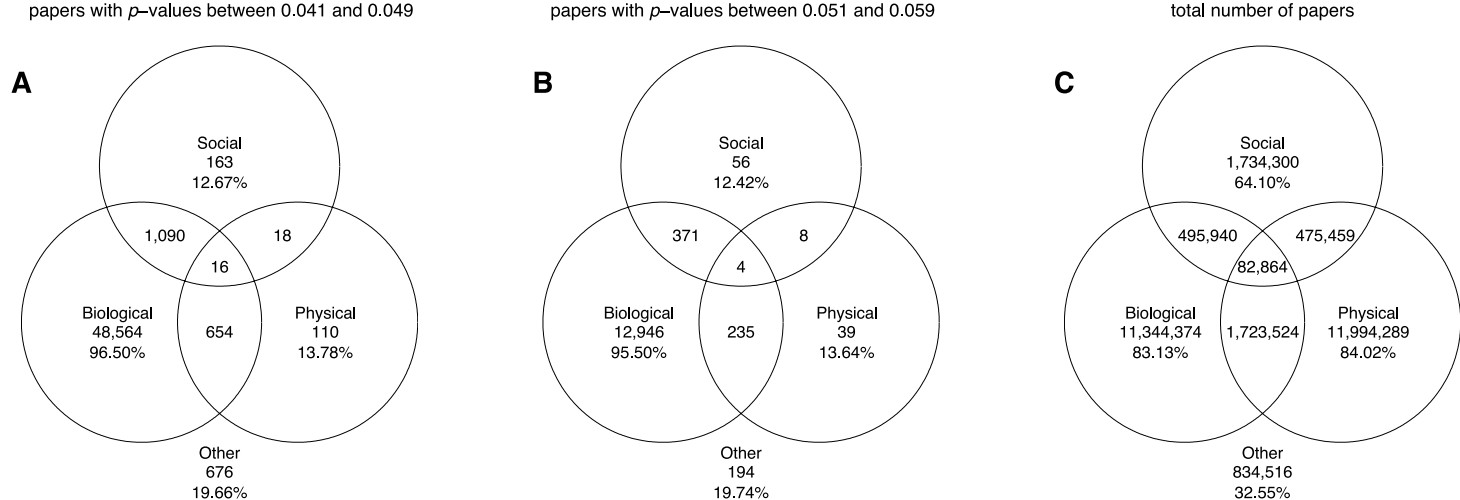

**Figure 14  Ratio of _p_ < 0.05 to _p_ > 0.05 per publication year, for three scientific disciplines.** The dashed lines represent the results of a simple linear regression analysis.

| papers with _p_-values between 0.041 and 0.049 | papers with _p_-values between 0.051 and 0.059 | total number of papers |

**A**

Social
163
12.67%

1,090          18

Biological          654          Physical
48,564                                110
96.50%                              13.78%

Other
676
19.66%

**B**

Social
56
12.42%

371          8

Biological          235          Physical
12,946                                39
95.50%                              13.64%

Other
194
19.74%

**C**

Social
1,734,300
64.10%

495,940          475,459

82,864

Biological          1,723,524          Physical
11,344,374                            11,994,289
83.13%                                84.02%

Other
834,516
32.55%

**Figure 15  Venn diagrams showing the numbers of papers reporting a _p_-value between 0.041 and 0.049 (A), the numbers of papers reporting a _p_-value between 0.051 and 0.059 (B), and the total number of papers (C).** "Other" refers to papers purely classified into subject areas outside the three disciplines. The percentages refer to the papers that were unique to each discipline (e.g., 96.50% of biological papers with _p_-values between 0.041 and 0.049 belonged purely to biological sciences).

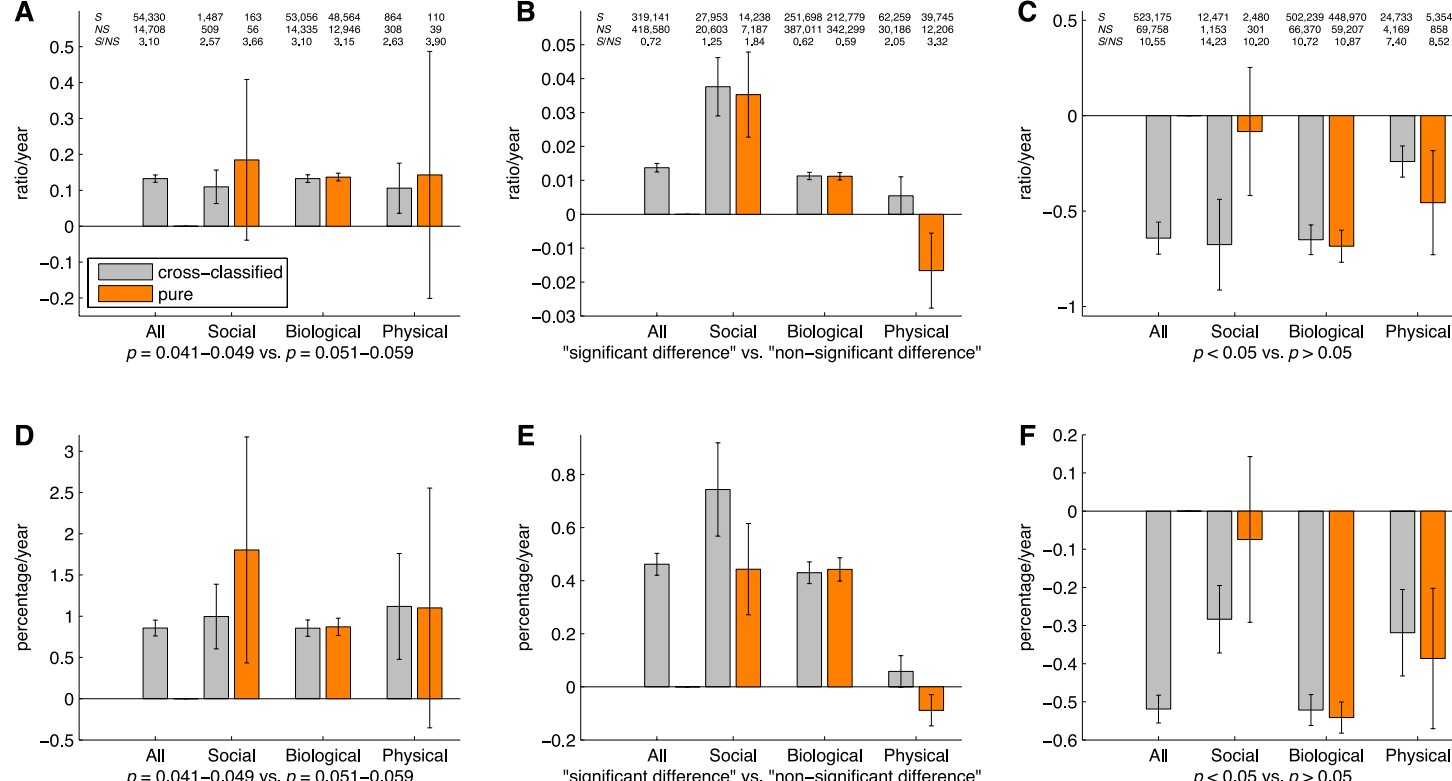

**Figure 16 Slope coefficients calculated using a simple linear regression analysis, for the ratios of significant (*S*) to non-significant (*NS*) results (*S/NS*; A, B, C) and the percentages of significant results (100%\**S*/[*S* + *NS*]; D, E, F).** The slope coefficients are reported for all papers, and for papers in three scientific disciplines, both for cross-classified papers (grey bars) and for pure disciplines (orange bars). The numbers at the top of the figure represent: (1) first row: number of papers between 1990 and 2013 reporting significant results (*S*); (2) second row: number of papers between 1990 and 2013 reporting non-significant results (*NS*); and (3) third row: ratio of significant to non-significant results (*S/NS*) calculated as the yearly *S/NS* averaged over 1990–2013. Error bars denote 95% confidence intervals.

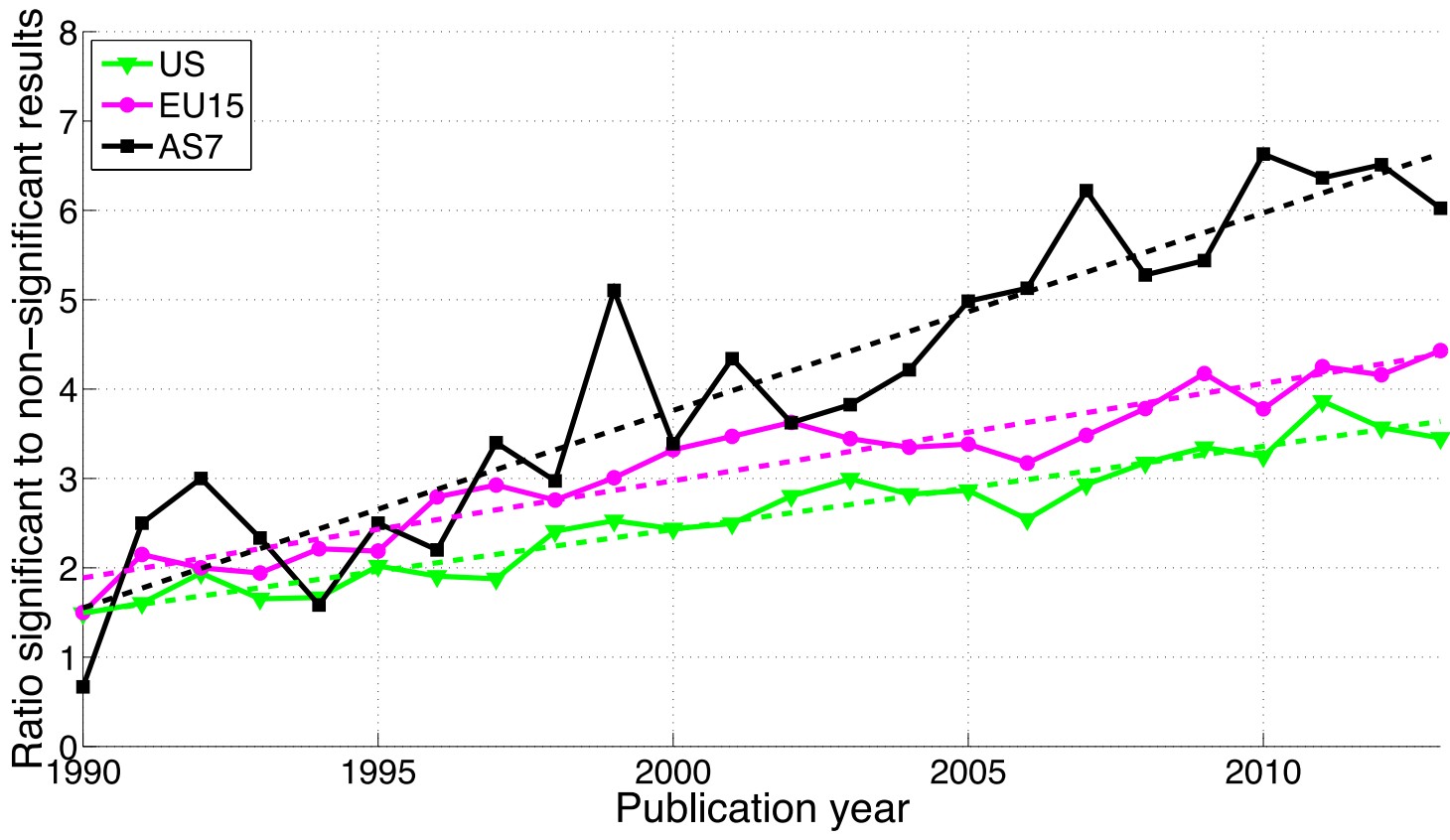

**Figure 17 Ratio of *p*-values between 0.041 and 0.049 to *p*-values between 0.051 and 0.059 per publication year, for three world regions.** The dashed lines represent the results of a simple linear regression analysis.

The virtual absence of *p*-values in the physical sciences can be explained by the fact that researchers in the physical sciences encounter low levels of sampling and measurement error and therefore do not need *p*-values. *Meehl (1978)*: "Multiple paths to estimating numerical point values ('consistency tests') are better, even if approximate with rough tolerances; and lacking this, ranges, orderings, second-order differences, curve peaks and valleys, and function forms should be used. Such methods are usual in developed sciences that seldom report statistical significance." Similarly, *Mulaik, Raju & Harshman (1997)* observed: "Another reason physicists often do not do significance tests is because they are not always testing hypotheses, but rather are trying to improve their estimates of physical constants." Eugene Wigner, who won the 1963 Nobel Prize in Physics for his contribution to the theory of atomic nuclei elementary particles, wrote: "if there were no phenomena which are independent of all but a manageably small set of conditions, physics would be impossible" (*Wigner, 1960*, see also *Hand, 2004*).

Overlap can also be seen between the social sciences and the biological sciences, with 86% of social sciences papers with *p*-values between 0.041 and 0.049 being also classified into biological sciences. This may not be surprising since psychology is often regarded as a "hub science" (*Cacioppo, 2007*). Summarizing, a hierarchy of sciences (if something like

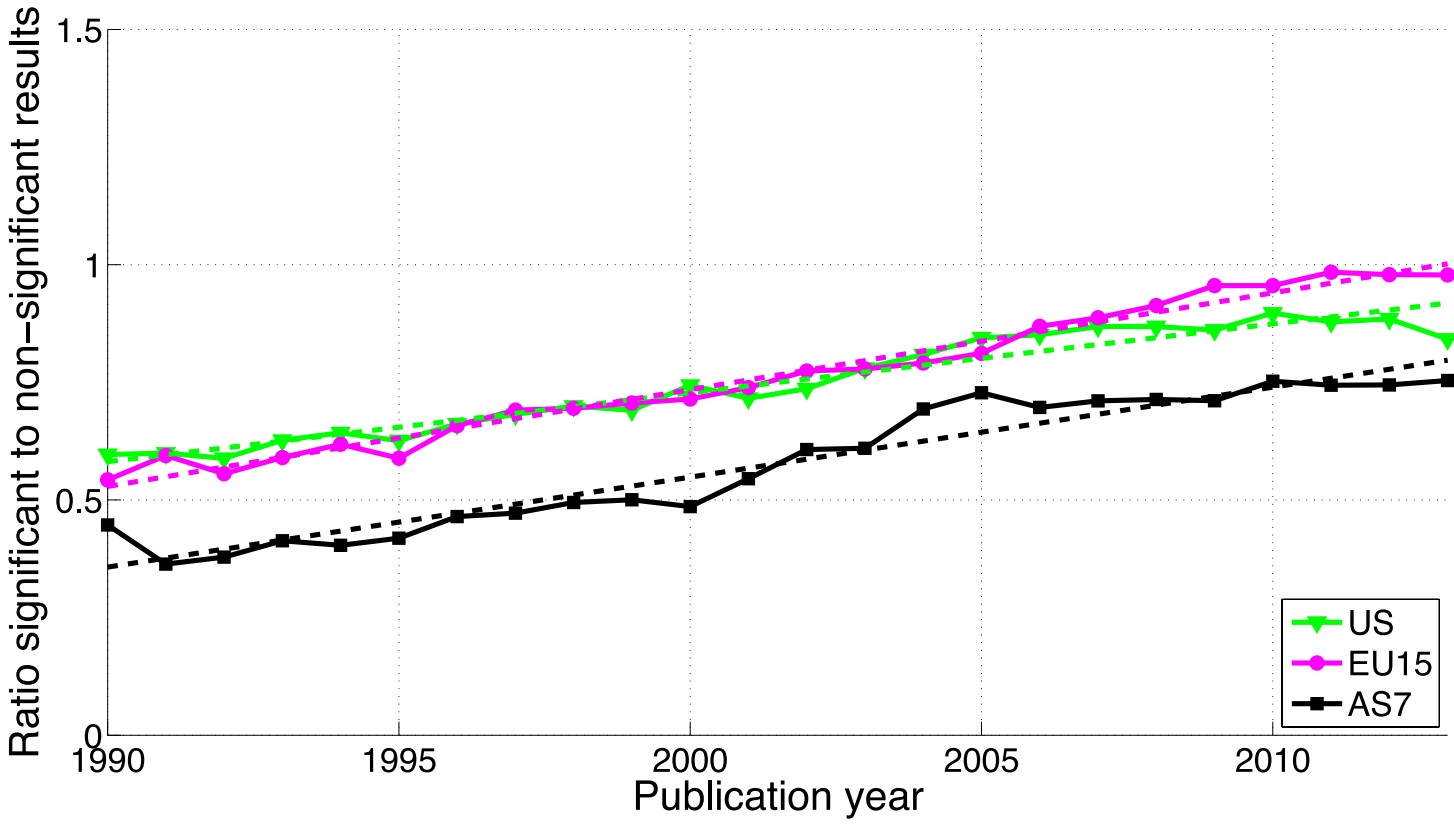

**Figure 18  Ratio of "significant difference" to "no significant difference" per publication year, for three world regions.** The dashed lines represent the results of a simple linear regression analysis.

that exists) is characterized by differences in the use of *p*-values in the first place, not by longitudinal trends of positive versus negative results.

We conducted longitudinal analyses for papers classified into multiple disciplines and for pure disciplines. The analyses of cross-classified and pure papers yielded similar results. We consider the results of the analyses in which the papers were cross-classified as more trustworthy than the pure analyses. Excluding all multi-disciplinary papers limits the sample size and leaves us with a core of classical mono-disciplinary journals that are not representative of all (multi-disciplinary) research.

## Comparisons of longitudinal trends between world regions

Previous studies have found that Asian research is more biased toward positive results than research elsewhere in the world (*Pan et al., 2005*; *Vickers et al., 1998*). Others have reported that it is the United States that overestimate effect sizes, especially in softer research (*Fanelli & Ioannidis, 2013*; but see *Nuijten et al., 2014*, questioning the robustness of the results of Fanelli & Ioannidis). Our analysis indicated that the ratios of significant to non-significant results and corresponding growth rates depend on the search term. For reporting $p < 0.05$ versus $p > 0.05$, we found that AS7 had the most rapid decline of the percentage of significant results, which is inconsistent with the hierarchy of sciences (*Fanelli, 2010*).

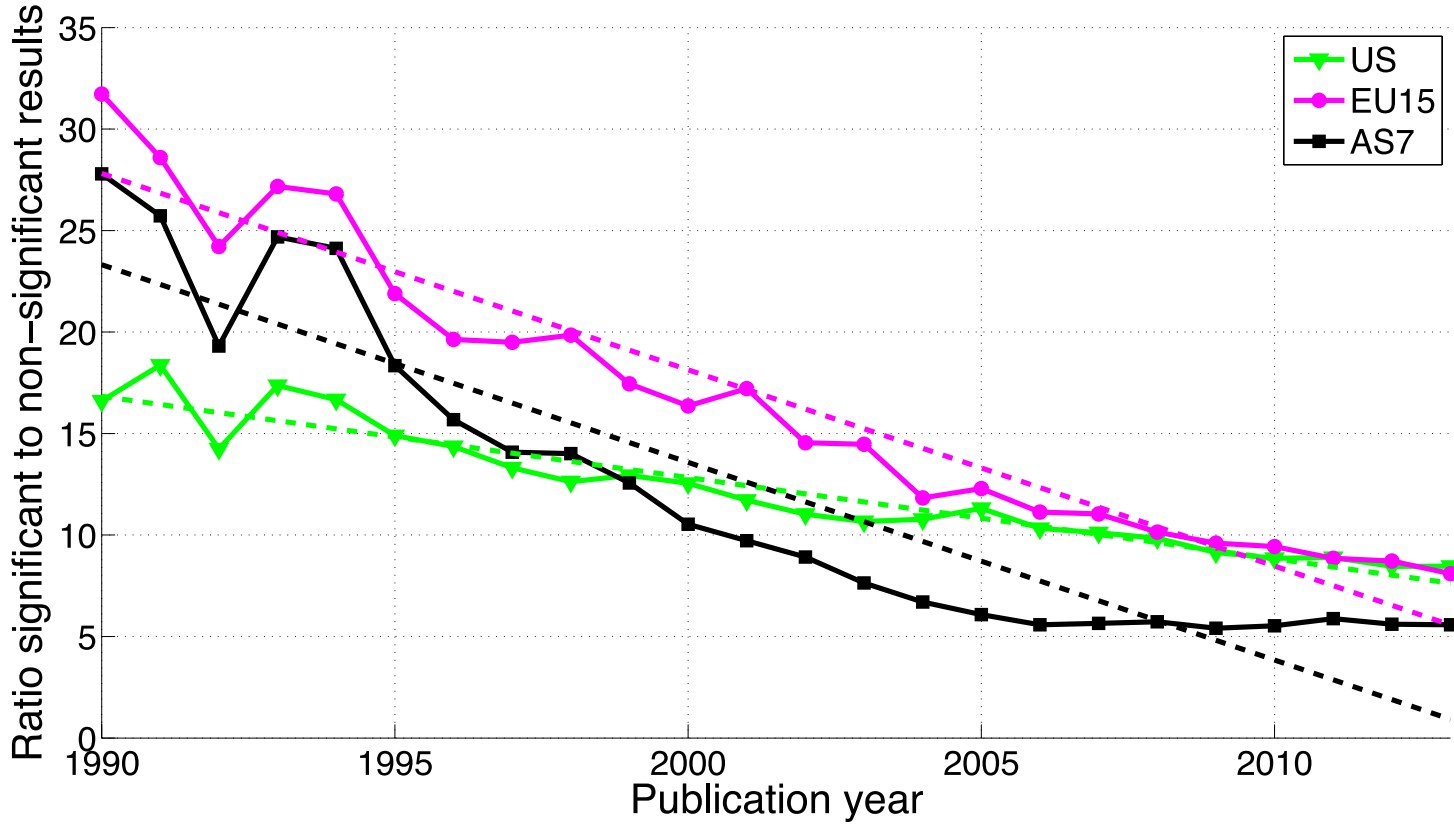

**Figure 19  Ratio of $p < 0.05$ to $p > 0.05$ per publication year for three world regions.** The dashed lines represent the results of a linear regression analysis.

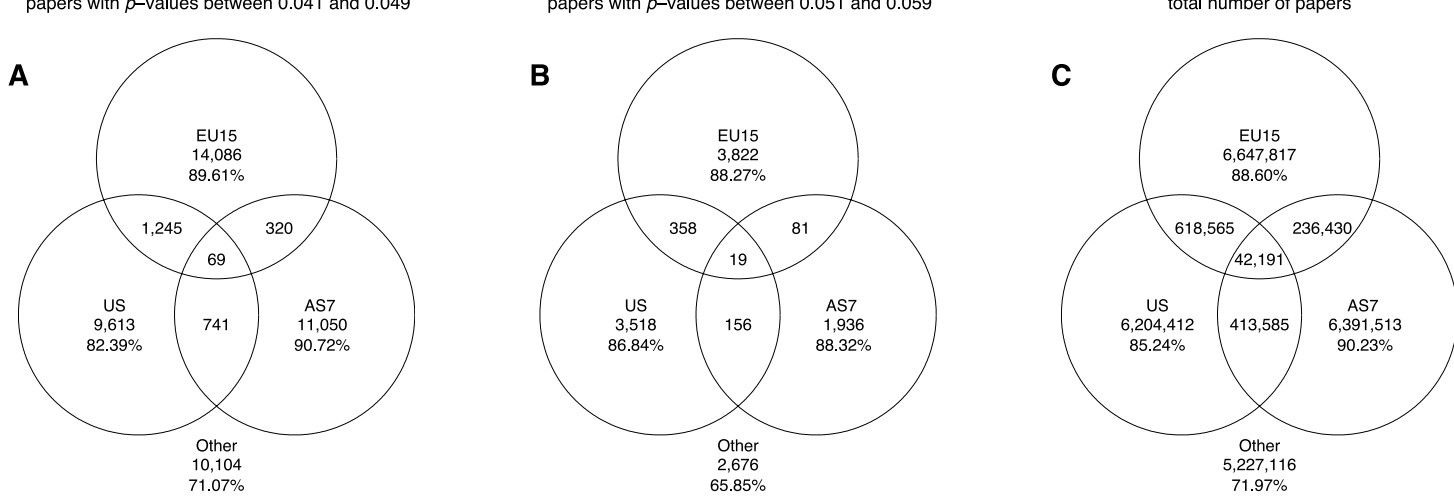

**Figure 20  Venn diagrams showing the numbers of papers reporting a $p$-value between 0.041 and 0.049 (A), the numbers of papers reporting a $p$-value between 0.051 and 0.059 (B), and the total number of papers (C).** "Other" refers to papers purely affiliated with countries outside the three world regions. The percentages refer to the papers that were unique to each world region.

de Winter and Dodou (2015), *PeerJ*, DOI 10.7717/peerj.733

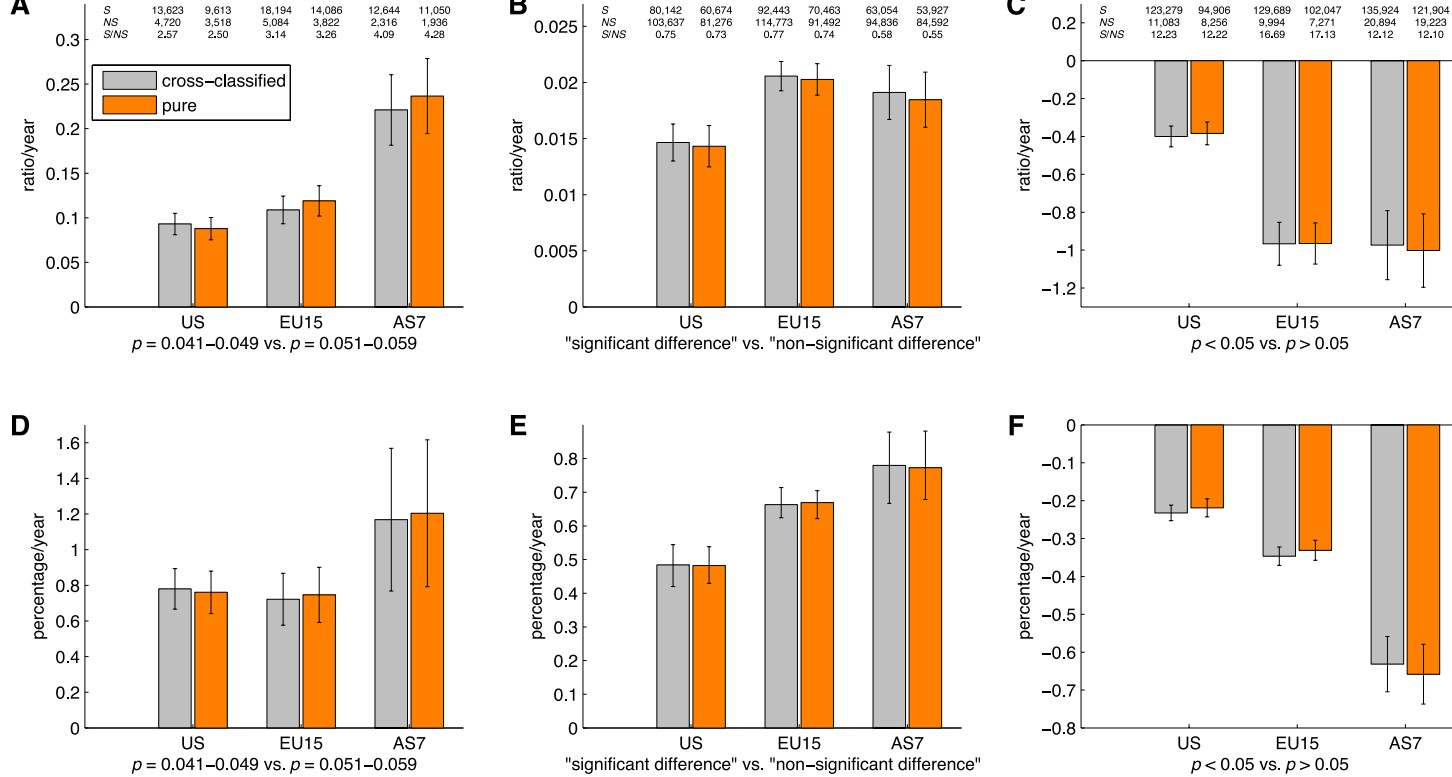

**Figure 21** **Slope coefficients calculated using a simple linear regression, for the ratios of significant (*S*) to non-significant (*NS*) results (*S/NS*; A, B, C) and the percentages of significant results (100%\**S*/[*S* + *NS*]; D, E, F).** The slope coefficients are reported for papers in three world regions, both for cross-classified papers (grey bars) and for pure world regions (orange bars). The numbers at the top of the figure represent: (1) first row: number of papers between 1990 and 2013 reporting significant results (*S*); (2) second row: number of papers between 1990 and 2013 reporting non-significant results (*NS*); and (3) third row: ratio of significant to non-significant results (*S/NS*) calculated as the yearly *S/NS* averaged over 1990–2013. Error bars denote 95% confidence intervals.

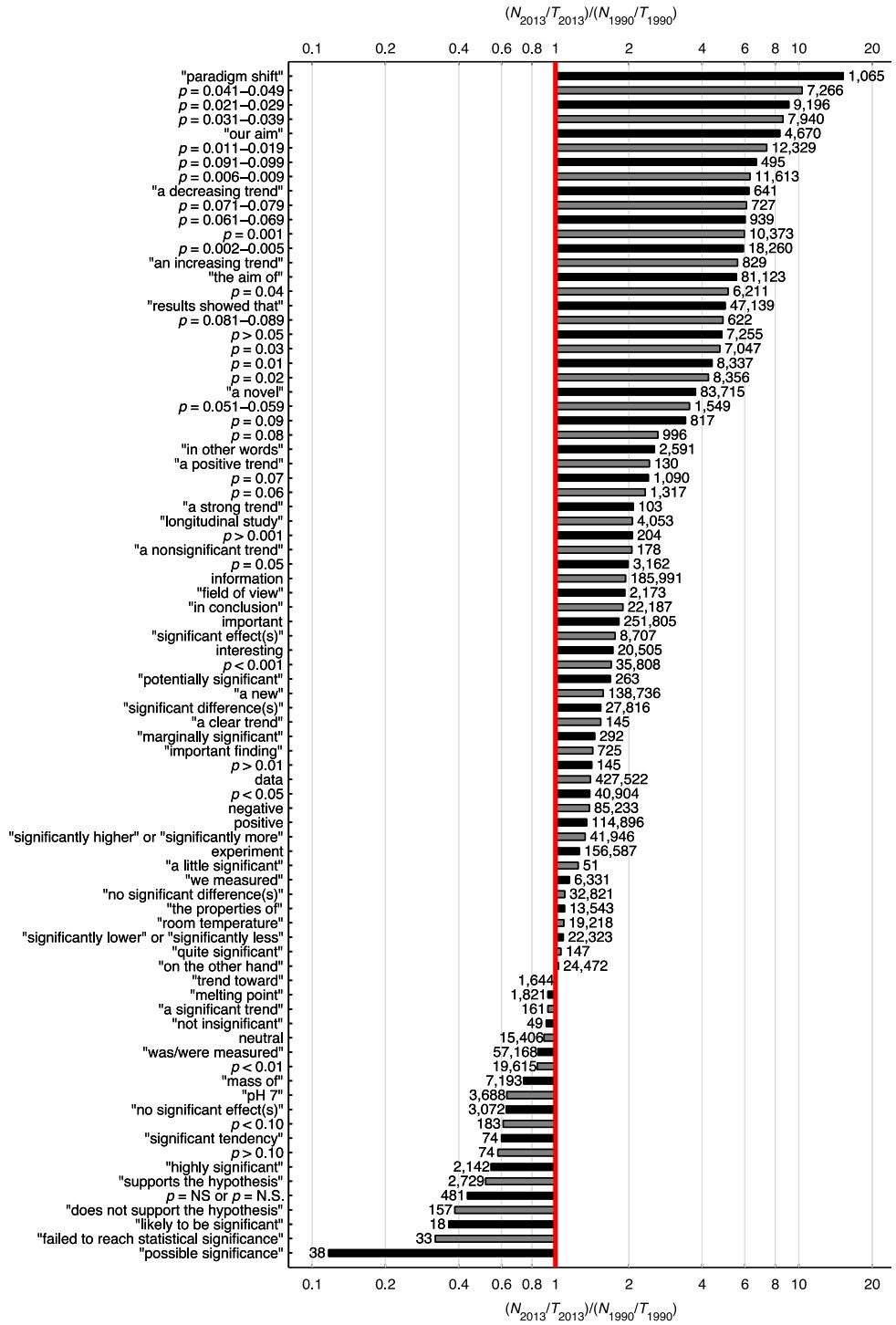

**Figure 22 Logarithmic plot of the 2013-to-1990 ratio** $(N_{2013}/T_{2013})/(N_{1990}/T_{1990})$**, where $N$ is the number of abstracts reporting a certain expression in 2013 or 1990, and $T$ is the total number of papers with an abstract in that year.** The number at the right end of each bar is $N_{2013}$. $T_{1990} = 561,516$ and $T_{2013} = 2,311,772$.

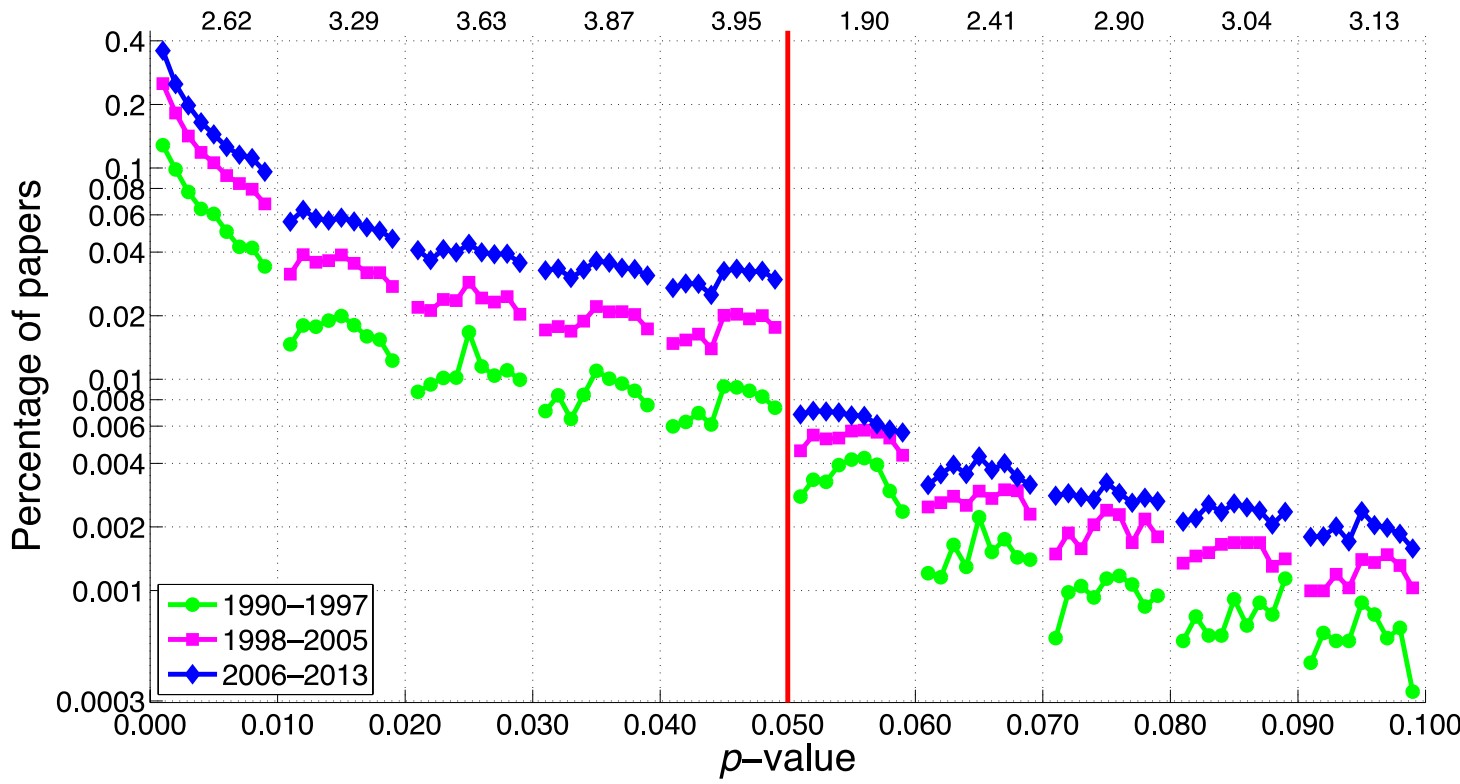

**Figure 23 Percentage of papers reporting a *p*-value as a function of the size of the *p*-value for three octennia.** The numbers at the top of the graph represent the ratio of the percentage of papers in 2006–2013 to the percentage of papers in 1990–1997 averaged across 0.001–0.009, 0.011–0.019, 0.021–0.029, etc.

We also found that researchers from Asia reported *p*-values between 0.041 and 0.049 at a high level compared to *p*-values between 0.051 and 0.059 (in agreement with *Fanelli, 2012*), but they were considerably *less* likely to use the phrase "significant difference" than the other two world regions (Fig. S3).

Although cross-national differences must exist (in the sense that any null hypothesis has to be false because there is always some effect), the size and direction of the effects are currently elusive. Regional differences in significance reporting are probably obscured by important moderators such as (1) the emergence of China as the second publishing power after the US during the last decade (*Leydesdorff & Wagner, 2009*), (2) that Asian researchers publish less frequently in the social sciences as compared to researchers from the US and EU15, and (3) that Asian researchers publish increasingly more over the years in the physical sciences (the least significance-oriented discipline) (see Fig. 2). To summarize: differences between world regions in the reporting of significance are too inconsistent to draw conclusions on cross-cultural differences in significance reporting.

## Limitations
### *Faulty inclusions*
Our automated string-search approach has some important limitations. First, our method is susceptible to faulty inclusions. To assess the occurrence of false positives that are

unrelated to hypothesis testing, we performed a small follow-up analysis. Using a random number generator, we selected 24 papers per discipline (one per year from 1990 to 2013) that reported a $p$-value between 0.041 and 0.049 in the abstract. Inspection of the 72 abstracts revealed one clear false positive, in the physical sciences (*Prandolini et al., 1998*; in which "$P = 0.048$" referred to a physical property rather than to a $p$-value). In the remaining 71 abstracts, $p$-values referred to a statistical test of some kind. However, the $p$-value was not always related to a clear-cut comparison between two or more groups or experimental conditions. For example, in some abstracts, the $p$-value belonged to a measure of association, to a variable contributing to a model, or to the assessment of model fit. Moreover, in our searches we made no distinction between unadjusted and adjusted $p$-values, such as those resulting from Bonferroni corrections for multiple testing. Abstracts often contain multiple $p$-values, some of which are overlapping or correlated, such as adjusted and unadjusted $p$-values of the same effect and $p$-values of subgroup analyses (*Ioannidis, 2014*). One strength of *Fanelli*'s (*2012*) work was that he manually examined the abstracts and/or full texts of the selected 4,656 papers testing a hypothesis. His manual checking of the content of each paper may have prevented faulty inclusions, albeit at the cost of a possible loss of objectivity.

### Literature misses

The papers that we assessed represent only a small fraction of all papers indexed in Scopus. For example, the papers with $p$-values between 0.041 and 0.049 and the papers with $p < 0.05$ correspond to respectively 0.18% and 1.7% of all 1990–2013 papers with an abstract indexed in Scopus. Moreover, our searches only focused on the abstract of the papers and by no means provide an exhaustive analysis of the semantic content of papers (note that *Fanelli*'s, *2012* sample was also generated by automatically searching for the sentence "test* the hypothes*" in the abstracts of papers included in the Essential Science Indicators database). Evidently, we have missed articles that report $p$-values only in the body of the article. We would argue that the abstract contains the main results and primary effects (see also *Pautasso, 2010*, who argues similarly), whereas many of the $p$-values that appear in the corpus of a paper are associated with secondary results. The corpus of a paper can contain a great number of $p$-values, for example associated with correlation matrices, which would not be desirable to include in the analysis. *Brodeur et al. (2013)*, who collected 50,078 statistical tests from the full text of 641 articles in Economics journals, reported that the average number of statistical tests per article is as high as 78 (with a corresponding median of 58). Some have warned that abstracts suffer from cherry picking of significant results (e.g., *Benjamini & Hechtlinger, 2014*; *Gøtzsche, 2006*; *Ioannidis, 2014*). According to *Gøtzsche (2006)*, "significant results in abstracts should generally be disbelieved." Similarly, *Ioannidis (2014)* argued that "P-values reported in the abstracts of published papers... are a highly distorted, highly select sample" and that "most investigators report some of the best-looking P-values in their abstracts." We argue that if selective reporting occurs in the abstracts, then focusing on the abstracts is *the* appropriate method to identify such behavior.

We focused our main analysis on alpha $= 0.05$, the most commonly used threshold of significance and the default in many statistical software packages. A full text in ScienceDirect for "$p < 0.05$ was considered" in 1990–2013 yielded 55,754 records, with the corresponding queries for $p < 0.01$ and $p < 0.001$ yielding only 1,221 and 299 records, respectively. In particle physics, statistical significance is typically reported in terms of 5 or 3 Sigma (*Hand, 2014*; *Lyons, 2013*; see also *Hasselman*'s *2014* commentary on a preprint of our paper). In genomics, typical alpha levels range between $10^{-4}$ and $10^{-8}$ (see Manhattan plots in genome-wide association studies; *Gibson, 2010*; *Gadbury & Allison, 2012*). Moreover, our search queries did not capture studies reporting of effect sizes with confidence intervals without *p*-values.

Scopus is known to be incomplete for publications prior to 1996 (*Elsevier, 2014*). We observed that the percentage of Scopus papers without an abstract dropped from 31.1% (254,326 out of 817,349 papers) in 1990 to 12.6% (334,988 out of 2,652,050) in 2013 (search query for papers without abstract: ALL({.}) AND NOT ABS({.}); search query for all papers in the database—with and without abstract: ALL({.})). To overcome this limitation and avoid artefacts due to an increased unavailability of abstracts for older papers, we focused our searches only on papers with an abstract available in Scopus.

A reviewer raised a concern about the completeness of the 2013 data in Scopus. According to the SCImago group that defines scientific indicators and rankings based on the Scopus database, "changes after October do not affect the database of the previous year any longer seriously" (*Leydesdorff, De Moya-Anegón & De Nooy, in press*). Considering that our data were extracted in November 2014, we expect that no major updates for 2013 are imminent and that the 2013 data are complete (i.e., as incomplete as the data from earlier years, in the sense that retrospective additions, such as old volumes of newly indexed journals, can always occur). During the last three months, we indeed noticed retrospective additions to the Scopus database, which suggests that small numerical changes are likely to occur in any future reproductions of this study. Specifically, between 28 August and 2 December, the number of papers with an abstract increased between 0.30% and 0.48% for papers published in 1990–1995, between 0.02% and 0.29% for papers published in 1996–2012, and with 0.43% for papers published in 2013.

### Literature representativeness

The three defined disciplines (social, biological, physical) are broad, and differences in statistical approaches between specialties within a discipline can be expected. As a follow-up analysis, we searched for *p*-values between 0.041 and 0.049 and *p*-values between 0.051 and 0.059 in three research fields of biological sciences: psychiatry, surgery, and cardiovascular medicine. A comparison between these three fields showed that the use of *p*-values between 0.041 and 0.049 were more prevalent in cardiovascular/heart journals, followed by surgery journals, and psychiatry journals (Fig. S5). We found no difference in growth rates between the three fields (Fig. S6). Of course, comparisons between other fields could have given different results.

### Automated versus manual searches

We opted for automated searches. Automated searches have been fiercely criticized in the past. *Ioannidis (2014)* stated that "the abstracts of the crème de la crème of the biomedical literature are a mess. No fancy informatics script can sort out that mess. One still needs to read the papers."

We argue that—provided that one uses a great number of diverse search terms and search strategies—an automated search should render a representative cross-sectional estimate of the trends in scientific publishing. Our manual inspection of the 72 abstracts mentioned in "Faulty inclusions" made it clear to us how difficult it really is to assess whether a paper supports a specific null hypothesis or not. We ran into three types of problems: (1) Issues of accessibility: Despite the fact that our university has 17,300 journal subscriptions, a large proportion of the full-text articles are hidden behind paywalls, which introduces serious availability bias (indicatively, for the 72 random abstracts, we did not have access to the full texts of 3/24 [13%] of papers in the biological sciences, 10/24 [42%] papers of papers in the physical sciences, and 10/24 [42%] of papers in the social sciences). (2) Issues of interpretation, especially when judging papers containing specialized terminology outside one's field. (3) The fact that many papers do not explicitly report on a hypothesis or might test multiple hypotheses at the same time. All three issues lead to problems of reproducibility. The results of an automated search, on the other hand, are highly reproducible.

For manual searches a sample size of 3,000 to 5,000 articles seems to be the maximum achieved so far (cf. $N = 4{,}656$ in *Fanelli, 2012*; $N = 3{,}701$ in *Leggett et al., 2013*; $N = 3{,}627$ in *Masicampo & Lalande, 2012*), and these relatively small sample sizes imply relatively low statistical power and reliability. *Masicampo & Lalande (2012)* investigated the distribution of $p$-values between 0.01 and 0.10 retrieved from 2007/2008 issues of three psychological journals, *Psychological Science* (*PS*), *Journal of Experimental Psychology: General* (*JEP*) and *Journal of Personality* and *Social Psychology* (*JPSP*). The $p$-values from *PS* were also re-collected and re-analyzed by a blind assessor, who found a distinctive peak of $p$-values just below the 0.05 threshold for this journal. Recently, *Lakens (in press)* re-analyzed *Masicampo & Lalande*'s (*2012*) datasets and concluded that the peak identified by the blind assessor, as well as the peak of $p$-values just below the 0.05 threshold found by *Leggett et al. (2013)* for *JPSP*, do "not replicate" and that "clearly, more data is needed." Prompted by Masicampo and Lalande' study, *Mathôt (2013)* automatically extracted 2,063 $p$-values associated with $t$-tests and ANOVA from the *Journal of Vision* and found no peak in the $p$-value distribution just below the 0.05 threshold. Our automated searches allowed us to retrieve 54,330 papers with $p$-values between 0.041 and 0.049, 14,708 papers with $p$-values between 0.051 and 0.059, 319,141 papers with "significant difference," 418,580 papers with "no significant difference," 523,175 papers with $p < 0.05$, and 69,758 papers with $p > 0.05$. Hence, sample size is an important advantage of automated searches.

## Conclusions and interpretations

We investigated longitudinal trends of positive versus negative results reported in the literature and compared these trends between scientific disciplines and between world regions. We found that both positive and negative results have become more prevalent and that the growth rates of both positive and negative results strongly depend on the search terms. The increase from 1990 to 2013 was evident for all $p$-value ranges, but strongest for $p$-values between 0.041 and 0.049.

The fact that $p$-values just below 0.05 exhibited the fastest increase among all $p$-value ranges we searched for suggests (but does not prove) that questionable research practices have increased over the past 25 years. This interpretation should be regarded with some caution, since a change of the $p$-value distribution can occur for various reasons, including changes in the file drawer effect and changes in the number of studies investigating a true effect (see also *Lakens, in press*).

In addition to questionable statistical practices, positive explanations for the observed trends are possible, particularly when one considers that not only significant but also non-significant results have increased. First, scientists may have become more knowledgeable, and therefore better able to formulate accurate predictions and design statistically powerful experiments that disprove a null hypothesis. Second, scientists may have become more likely to use, and report the results of, statistical significance testing. The observed longitudinal trends of the reporting of $p$-values might thus occur because of science becoming more empirical, organized, and quantitative. Structured reporting in abstracts is getting preferred (and recommended in journal instructions for authors) over a narrative summary of research findings (see *Ripple et al., 2011*, who analyzed about 5.5 million MEDLINE records and found that the frequency of structured abstracts increased from 2.5% in 1992 to 20.3% in 2005). The growing popularity of expressions such as "the aim of," "our aim," and "results showed that" is consistent with the notion that structured reporting in abstracts has been embraced by journals. Moreover, as reminded by *Low (2014)* in a reaction to the preprint of our paper, the increase of $p$-values may be associated with the fact that people nowadays do not use lookup tables, but report numeric $p$-values instead (*Sterne, 2002*). Third, the observed increase of negative results (Fig. 3) may be caused by a growing replication movement providing a counterforce to positive results (cf. mega-journals such as PLoS ONE which judges papers based on their technical soundness rather than statistical significance; *Binfield, 2009*), and the increasing use of $p$-values might be a manifestation of dynamic exchange of information within the scientific network. Besides, as indicated by *Popper (1959)*, theories are "*in an asymmetrical way, falsifiable only*: they are statements which are tested by being submitted to systematic attempts to falsify them" (italics in the original). In this line, a single falsification can be worth more than numerous confirmations of an established null hypothesis. So, perhaps some asymmetry between positives and negatives results in innovative research fields is expectable, or even desirable.

### Funding

The authors declare there was no funding for this work.

### Competing Interests

The authors declare there are no competing interests.

### Author Contributions

- Joost CF de Winter conceived and designed the experiments, performed the experiments, analyzed the data, contributed reagents/materials/analysis tools, wrote the paper, prepared figures and/or tables, reviewed drafts of the paper.
- Dimitra Dodou performed the experiments, prepared figures and/or tables, reviewed drafts of the paper.

### Supplemental Information

Supplemental information for this article can be found online at http://dx.doi.org/10.7717/peerj.733#supplemental-information.

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
