# Peer review of "A surge of p-values between 0.041 and 0.049 in recent decades (but negative results are increasing rapidly too)"

_PeerJ, doi:10.7717/peerj.733_

## Round 0.1 · original submission · Major Revisions

In addition to the feedback from two peer-reviewers on your submitted manuscript, this editorial decision was also informed by valuable feedback provided by three scholars commenting on the PeerJ Pre-Print that you also submitted.

The way in which the data were collected is a serious shortcoming of this study that requires additional attention to ensure the validity of the findings. While caveats were included at the end of the paper, they were not given sufficient prominence. The methodological issues undermine the comparative analysis and interpretation. The discussion section seems particularly speculative in light of the experimental design limitations.

Overall, the basic reporting is fine. There are a few problems that could be easily corrected.

1. Please revise any figures for accurate reporting.
a. The Y-axis label and caption seem to be contradictory for Figure 1, as one reviewer noted. Is this the number of papers or percentage of papers?
b. Figure 2 is potentially misleading/confusing in its illustration of partial data for the most recent year. This could be minimized by either omitting the partial year data from the graph or projecting the trend to the end of the year.
2. One of your references is out of alphabetical order (Ferguson and Heene 2012, which appears before Fanelli and Ionnidis 2013).
3. A reviewer suggested, and I agree, that your methods and findings regarding the textual reporting trends deserve inclusion in the main text rather than reserved to supplementary material.
4. It is a bit awkward and misleading (and potentially false) throughout the manuscript when you discuss “positive” versus “negative” results when in actuality the data, as you indicate, are p-value reporting between 0.40-0.49 and 0.51-0.60 and the textual reporting of absence/presence of statistical significance.
5. Always read your manuscript from start to finish for grammatical problems.

There are a number of features of the experimental design that warrant additional justification or require additional research to ensure the validity of the findings:

1. The use of Scopus may have introduced coverage bias and selection bias.
a. On what basis did you make the decision that Scopus was superior to Google Scholar, Web of Science, or other options? The rationale should be provided. What made Scopus “most accurate and powerful”?
b. You indicated that Scopus allows cross-categorization (a paper can be assigned to more than one subject area and, accordingly, to more than one category you created). Please elaborate on how this affects your analysis. One of the Pre-Print comments (Fanelli) was particularly critical of this undermining any categorical comparisons you make with such overlapping data.

2. The sampling strategy (search strings) used could have introduced extensive selection bias. The search strings used for data collection seem too narrow/strict to support your subsequent analysis and interpretations. As you acknowledged in your response to de Ruiter’s PrePrint Comment, “our search strings catch only a small portion of papers”) and emphasize that’s why you discuss the results in terms of a ratio of positive to negative p-value reporting. It is necessary for a revised manuscript to provide sufficient justification for the choice of search strings used (and variant search strings omitted) and explain how these limitations do not bias and invalidate the results or otherwise frustrate the interpretations of those results. Here are three issues that you should consider:

a. Please address why your p-value search was restricted to the range of 0.040-0.049 and 0.051-0.060, which neglects tests of significance that may have been performed (and reported) at other alpha levels (the p-value cut-off decision for determining what is or is not significant is always somewhat arbitrary) and neglects reports of significance with adjusted p-values (e.g. such as those resulting from Bonferoni corrections for multiple testing) outside that range. It may be relevant for readers for you to clarify also why you omitted a third query of reports of exactly p=0.05, p>0.05, and p<0.05 from the main analysis. I was surprised (as was one of the reviewers) when I got to page 11 to learn you did perform a “supplementary analysis” looking at reports of p<0.05 and p>0.05. That this “supplementary analysis” gave you a dramatically different picture than your main analysis is not surprising, however. It speaks to the weaknesses of the experimental design and calls the validity of the findings from your main analysis into question.

b. You buried a note on the bottom of page 10 that your experimental design is limited because you did not verify whether the p-values were in papers in which authors were testing a hypothesis. Your approach and discussion assumes p=0.05 is the threshold for statistical significance in all of the sampled publications. Why does your assumption not undermine the data collection, results, and subsequent interpretations? In a few places in the main text, you discuss findings as “marginally significant” and “marginally non-significant.” I draw your attention, which might provide some insight for you, to a blog post by Professor Jim Wood at The Mermaid’s Tale on May 13, 2013 titled “It’s time we killed off NHST,” Accessible at http://ecodevoevo.blogspot.com/2013/05/lets-abandon-significance-tests.html. Given that you (not necessarily the authors) attribute “significant results” and “not significant results” to any p-value reported between the two ranges without performing content analysis of the papers’ full texts, how can you ensure the validity of that judgment call? It seems fundamental to your subsequent comparative analysis.

c. Please also address why your textual search was restricted to “difference” (not capturing papers that may have used “different”) and why you did not attempt to search for additional qualitative descriptors (such as “higher,” “brighter,” “stronger,” “more,” “lower”, “fewer”, “less”, etc.). Why were more variants (e.g. “significantly different”, “differed significantly”, “did not differ significantly”) omitted from your sampling strategy? Your explanation that you were attempting to replicate Pautasso (2010) was unsatisfying. Authors might write that “the differences were not statistically significant” rather than “there was not a significant difference,” but your search strategy would only identify the latter. This introduces systematic bias in your dataset. Accordingly, any trends found in your longitudinal data (such as those shown in Figures 7 and 8) may just be artifacts from coverage and selection biases rather than indications of meaningful differences and changes in textual reporting done in the various disciplines studied over that period.


3. Your approach uses an automated analysis of abstracts rather than manual content analysis of the full texts of the papers identified with your search strings. There are two issues there: (1) collected data from abstracts rather than the main text of the publication and (2) automated analysis rather than manual analysis. Please elaborate on the justification for your approach with regard to both factors. How confident are you that your strategy did not systematically miss articles in which the authors do not report p-values in the abstract but do report them in the main text? How would this affect the conclusions and speculations that you drew from your findings? While automation is less subjective than manual analysis, it is erroneous to suggest it is “free” from human biases, as coding bias is not the only type of bias. Automation provides speed and precision but can come at an expensive sacrifice to accuracy. The richness of the data can be lost when context has not been adequately taken into account. You noted on page 10, that your approach did not even verify whether reported p-values related to hypothesis testing. Accordingly, interpretations of the data are problematic and largely speculative.

The paper addresses an interesting and important topic, but PeerJ focuses on the soundness of the science (basic reporting, experimental design, and validity of the findings). I would encourage you to reconsider the experimental design to overcome some of its current limitations and to verify the robusticity of the data. I would look forward to seeing your revised work.

·

Basic reporting

Many of the graphs report "Number of papers (%)" but these are different terms. What is actually reported is the % not the number of papers. In addition, the captions of these graphs indicate that what is reported is the proportion of papers, but I think it is actually %.

The text should probably reference:
Masicampo EJ & Lalande DR (2012) A peculiar prevalence of p values just below .05. Quarterly Journal of Experimental Psychology, 65(11): 2271-2279.

Experimental design

I found it strange the the searches for p-values did not use {p=0.04} OR {p=.04} OR {p=0.05} OR {p=.05}. Surely not everyone writes the p values out to 3 decimal places.

Validity of the findings

The reference to an excess of positive results or excess significance (in various places) seems unwarranted (or at least unfounded in the current text). The authors do not know if there is an excess of significance based on their data.

Additional comments

I'm not sure going through abstracts is the way to really understand these issues because in most studies the key information is in the main text rather than in the abstract. Nevertheless, the method and results in this paper add to the existing literature. The observation that different methods give different conclusions (e.g., regarding the increase or decrease in negative results) is important for people trying to use or interpret these types of studies.

The manuscript includes a number of typos and small errors. I have marked up a PDF copy and attached it.

·

Basic reporting

No comments

Experimental design

No comments. See below.

Validity of the findings

No Comments -see below.

Additional comments

I found it hard to Identify which box my comments belong in. I’m putting them all here.
1. P2: the authors say “it is unclear why Fanelli reported an over 22% increase in the abstract” but 86/70 is an increase of 22.9%.
2. P4: please clarify the relevance of the remark “Note that a paper can belong to more than one subject area”. Does that mean that the data in the different groups are not wholly independent? This is clearly the case of regions too. What is the impact of this overlap? That isn’t mentioned in the Discussion.
3. It seems that no attempt was made to cross-relate multiple P values appearing in a single abstract. It would be interesting to know about the cross-occurrence of P-values in the two ranges studied, and also to know if authors are more likely to give some P values greater than 0.05 when they also report some P values <0.05 (not necessarily within the narrow range 0.04-0.049).
4. Likewise no attempt was made to identify whether papers were indeed reporting research findings. Maybe fewer non-research articles are published? Indeed this seems likely give the growth of new journals that largely publish only research findings. Further, there are several research areas within the life sciences where hypothesis testing isn’t relevant - one example is diagnostic test accuracy studies which estimate sensitivity and specificity of a test.
5. Can the authors comment on the delay between publication and indexing on Scopus. Specifically, are the data for 2013 expected to be complete? Also, presumably Scopus has increased its coverage of journals over the years of this study, not least because so many new journals have started up. Presumably this won't affect results expressed as %s.
6. I think it would be helpful to invert the small %s and observe that in 1990 1 paper in 500 reported a P value in the abstract between 0.051 and 0.060 whereas in 2013 it was about 1 in 300. (Of course you can use 2014 but this estimate is less precise, and I’m not convinced that the large change from 2013 to 2014 is genuine.)
7. More than once the authors draw inferences from overlapping or non-overlapping confidence intervals. While non-overlapping 95% CIs do indicate P<0.05 (indeed somewhat less than 0.05), overlapping confidence intervals may also correspond to P<0.05. In fact such comparisons/tests aren’t strictly appropriate as the same articles are counted multiple times. (It’s unclear to what extent – and this could usefully be reported.)
8. P9: The Discussion opens with the following observation: “We investigated longitudinal trends of positive versus negative results reported in abstracts and compared these trends between disciplines and between world regions.” The mention of positive versus negative results isn’t completely accurate. In particular, the authors excluded P values outside the range 0.04 to 0.06. I think it’s a pity that they haven't done similar analyses of small P values and those with P>0.06. Also, of course, many abstracts will include statements such as P<0.05 or P>0.05 or the appalling P=NS (14000 of these on PubMed!).
9. The results are clear. The interpretation is tricky though. First, as noted, only a subset of P values was collected. Second, there may well have been major shifts in the mix of different research types over 25 years. I don’t think that the trends seen can be explained by for example more studies of types that don’t need P values, but I do think it could be a contributor to the trends seen. Also the interpretation is strongly affected by the restrictions in the available information which had to meet specific formatting. The strength of the study – the huge sample size – is thus somewhat offset by the automated and limited nature of the data that were extracted, making it impossible to study some more interesting aspects. Also my experience is that within clinical research, itself only part of the life sciences literature, there are large difference s in statistical approaches between specialties e.g. between psychiatry, surgery, and cardiovascular medicine. It is likely that there is variation in the extent of selective reporting of significant results across such fields, which doesn’t have a chance of being detected in this study.
10. I was surprised to learn part way through the discussion that the authors had indeed looked at some smaller P values; specifically they studied P values of 0.012, 0.022, 0.032, and 0.042. This material is in an appendix. The Methods section should at least outline all the analyses that were done. They also looked at “p < 0.05” versus “p > 0.05 – gain I find this information worthy of inclusion in methods and summary in results, not just in the Discussion. That said I do wonder if some many analyses are merited even if in supplementary material.
11. A specific result may be reported with a P value or not, and may be described as significant or not. I'd expect that there was an association between these two ways of presenting findings. The comparison of these two types of information is rather unsatisfactory. Also note that there is only a partial overlap between the two classifications because of the restricted range of P values examined.
12. The authors discuss several limitations. They might add that there are likely to be shifts in both the way in which research is done and reported, and within the latter changes in the way researchers write abstracts. For example, there has been a shift in most filed from unstructured to structured abstracts, certainly in the medical field, which might be associated with different practices in reporting numerical results.
13. Overall I feel that the Discussion is long and speculative; it fails adequately to recognise all of the weaknesses I have mentioned.
14. Abstract: I find this a strange idea: “it is unclear whether negative results are disappearing from papers.” At least it isn’t a question I can understand. This sentence joins two unrelated ideas. The text on p3 is much clearer: “However, it is unclear whether the prevalence of negative results is decreasing over time, whether the increase of positive results is stronger in the softer disciplines (i.e., social sciences) as compared to the harder disciplines (i.e., physical sciences), and whether different regions in the world exhibit different tendencies in reporting positive versus negative results.”

---

## Round 0.2 · accepted · Accept

I appreciate your thoughtful attention to the issues raised in the previous editorial decision and raised by the two reviewers of the original submission. Incorporating extensive discussion of the limitations of your methodology as well as conducting additional searches for textual variations of reporting statistical significance were very important improvements to the work. Please work closely with the PeerJ staff to ensure that the figures are correctly displayed.

·

Basic reporting

No comments

Experimental design

No comments

Validity of the findings

No comments

Additional comments

My relatively minor concerns with the original submission have been sufficiently addressed in the revision.

The only issue to watch out for is the figures. All the figures in the main text appear to be cut off on the right side. Some figures are also missing regression lines, the venn diagrams are not outlined, and bar plots are missing their bars. The figures at the end of the manuscript appear to be proper, so my guess is that the problems are peculiar to the generated PDF.

·

Basic reporting

.

Experimental design

.

Validity of the findings

.

Additional comments

This paper represents a mammoth investigation of P values in abstracts of scientific articles. Many comments were made in the first round of reviews. The authors’ responses to these are highly satisfactory, and the paper is much improved. I wouldn’t have done the study this way but the methods are clear and the information provided is fascinating, not least when the different scientific disciplines are compared.

I have not reread the whole extensive updated manuscript but am happy with the parts I have examined, especially in relation to the yellow highlighted edited passages.

I have a few small points
Trends seen over time in the reported P values will necessarily represent a mix of several trends, including trends in the actual results of studies, trends in data manipulation practices, and trends in the way that authors write abstracts. For example if abstracts are getting longer there is room to report more numerical findings. I suspect too there has been a trend (over decades) away from qualitative towards quantitative reporting of findings. . Interpretation of time trends as due to just one specific cause is thus unwise.
P3: “Pautasso (2010) further showed that the positive results had increased more rapidly than the negative results, hence yielding a “worsening file drawer problem”,” - this assumes that there is a fixed % of significant results over time, which is a questionable assumption.
P3: Sacket should be Sackett I presume, but I can’t check as this reference isn’t given. Maybe other citations are missing?